# One Hundred Neural Networks and Brains Watching Videos: Lessons from Alignment

**Christina Sartzetaki**[1], **Gemma Roig**[2], **Cees G.M. Snoek**[1] **& Iris I.A. Groen**[1]
[1]Informatics Institute, University of Amsterdam, The Netherlands
[2]Department of Computer Science, Goethe University Frankfurt, Germany
{c.sartzetaki, i.i.a.groen}@uva.nl

## Abstract

What can we learn from comparing video models to human brains, arguably the most efficient and effective video processing systems in existence? Our work takes a step towards answering this question by performing the first large-scale benchmarking of deep video models on representational alignment to the human brain, using publicly available models and a recently released video brain imaging (fMRI) dataset. We disentangle four factors of variation in the models (temporal modeling, classification task, architecture, and training dataset) that affect alignment to the brain, which we measure by conducting Representational Similarity Analysis across multiple brain regions and model layers. We show that temporal modeling is key for alignment to brain regions involved in early visual processing, while a relevant classification task is key for alignment to higher-level regions. Moreover, we identify clear differences between the brain scoring patterns across layers of CNNs and Transformers, and reveal how training dataset biases transfer to alignment with functionally selective brain areas. Additionally, we uncover a negative correlation of computational complexity to brain alignment. Measuring a total of 99 neural networks and 10 human brains watching videos, we aim to forge a path that widens our understanding of temporal and semantic video representations in brains and machines, ideally leading towards more efficient video models and more mechanistic explanations of processing in the human brain.

## 1 Introduction

Humans are extremely efficient in processing the constant streams of visual information they receive, relying on motion and temporal information on top of visual semantics to understand their environment (Sekuler et al., 2002). How current state-of-the-art video models compare to that standard is a question that is often addressed by comparing their performance to human baselines (Zhou et al., 2018a; Andonian et al., 2020), but is much more under-explored with respect to internal representations. Representational alignment was defined in Sucholutsky et al. (2023) as "the extent to which the internal representations of two or more information processing systems agree". It is a cornerstone of cognitive computational neuroscience (Kriegeskorte & Douglas, 2018), a discipline that aims to identify neural mechanisms underlying cognition by employing task-performing computational models, such as deep neural networks, for hypothesis testing. This has been shown to be a highly progressive research program yielding novel neuroscientific insights (Doerig et al., 2023). Identifying model design choices that strongly impact alignment can not only shed light on the underlying brain mechanisms, but also guide machine learning on borrowing brain's advantages such as efficiency and robustness, for example using brain-aligned designs as starting points for further model development (Lu et al., 2024b). Our work aims to fill the gap in exploring the representational alignment of deep video models to the human brain, joining and increasing understanding of temporal modeling in brains and machines, and pointing towards more efficient video models.

Human visual information processing has classically been separated into two separate streams, ventral and dorsal (Mishkin et al., 1983). The ventral stream is thought to process more static information and the dorsal stream more dynamic information (Kravitz et al., 2011), whereby a motion area MT (Culham et al., 2001) has also been identified. More recent work found evidence for a third visual pathway specializing in dynamic social perception (Pitcher & Ungerleider, 2021; Küçük et al., 2024). Another organizing principle that has been identified in human visual processing is a hierarchy of temporal receptive fields, with temporal integration extending over longer time-scales in higher compared to lower visual areas (Hasson et al., 2008; Zhou et al., 2018b; Groen et al., 2022;

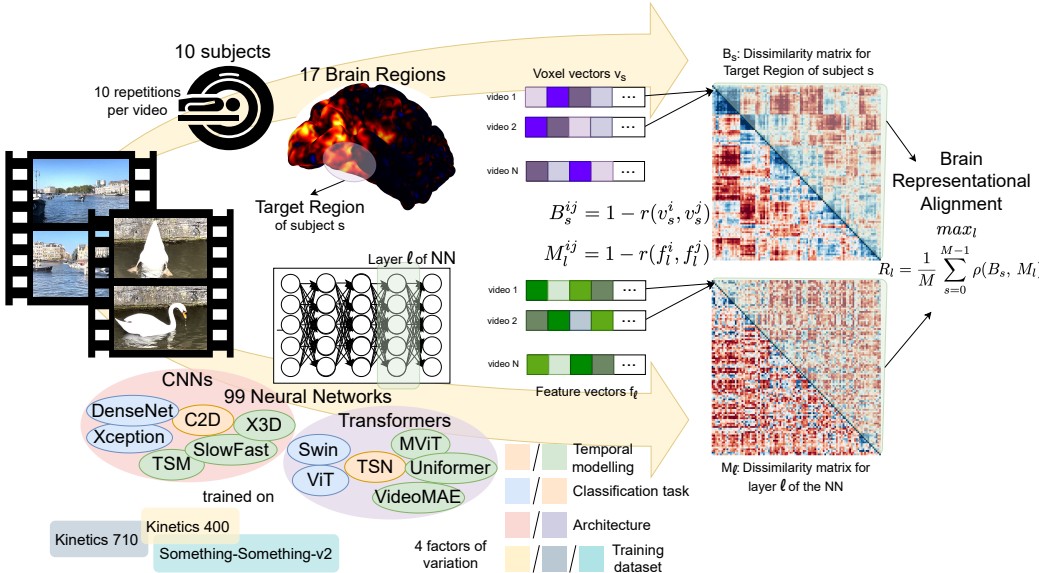

Figure 1: Overview of our pipeline for benchmarking video models on brain representational alignment. We systematically compare 99 neural networks across 4 factors of variation on their alignment with fine-grained brain regions measured with Representational Similarity Analysis.

Brands et al., 2024). These neuroscience findings collectively suggest the presence of specialized mechanisms for motion and temporal information processing in the human brain.

In video AI, specifically machine learning, temporal information is usually modeled as a third input dimension, using mechanisms such as 3D convolutions (Tran et al., 2015; Carreira & Zisserman, 2017); however there are only limited similarities with processing in the brain, as the effective temporal receptive fields in these networks usually stay constant throughout processing layers. Older models handling RGB images and optical flow as separate streams (Simonyan & Zisserman, 2014) bear some resemblance to dual-stream processing in the brain, but most successful designs fall far from direct comparison (Feichtenhofer et al., 2019; Lin et al., 2019). Recent papers focus on video Transformers (Liu et al., 2022; Wang et al., 2023b), including hybrids to combine benefits of CNNs and Transformers (local structure and long term dependencies) in temporal modeling (Li et al., 2022; 2023a). Comparing the variety in engineering solutions from video-AI to the way humans process video can help adjudicate between models in terms of their ability to capture the temporal aspects of information processing in the brain and increase understanding on the representations they compute.

A line of work has recently set path in large-scale systematic benchmarking of models on representational alignment, to both the human brain and behavior (Conwell et al., 2022; 2024; Muttenthaler et al., 2023), advancing the understanding of how these models internally compare to humans on a common basis. However, these works have so far been limited to the domain of static images; a comprehensive analysis of video model alignment with human video processing is still lacking.

**Our contributions:**

- We perform the first large scale benchmarking of video models on human brain representational alignment, on a recent publicly released video fMRI dataset. Specifically, we consider 47 video models for action recognition and 41+11 image models for object and action recognition.
- We decouple the alignment effects of temporal modeling from those of action space optimization by adding image action recognition models as control, as well as examine the impact of model architecture and training dataset, all comparing across a fine-grained variety of brain regions.
- We show that temporal modeling is the important factor for representational alignment to early brain regions, while action space optimization for alignment to late brain regions. We find distinct patterns emerging in different processing stages of CNNs and Transformers, along with an effect of training biases on alignment to functionally selective areas.
- We report a significant negative correlation of model FLOPs to alignment in several high-level brain areas, indicating that computationally efficient neural networks can potentially produce more human-like semantic representations.

## 2 RELATED WORK

**Large scale brain datasets.** Compared to artificial intelligence, the field of computational cognitive neuroscience has been exposed to the concept of benchmarking only recently (Schrimpf et al., 2020), with the majority of prior works consisting of stand-alone assessments of representational alignment on their own brain data for one or only a few models. Works such as BrainScore (Schrimpf et al., 2018) and the Algonauts challenges (Cichy et al., 2019; 2021; Gifford et al., 2023) have set the stage for representational alignment research on benchmark datasets where any model can be compared and comparisons can be widely accessed by everyone in the community, generating new insights via data-driven exploration and field-mapping. Datasets such as NSD (Allen et al., 2022) and THINGS (Hebart et al., 2023) have been established as benchmark image stimuli brain imaging datasets due to their size and high signal quality. For video stimuli there has also been some recent development (Dima et al., 2022; Zhou et al., 2023; McMahon et al., 2023), that is mostly focused on social actions. Most recently the Bold Moments Dataset (BMD) (Lahner et al., 2024) was introduced, putting forth a large-scale, highly reliable video fMRI dataset with extensive quality control to ensure suitability for AI model comparisons. *In this work,* we conduct the first extensive benchmarking of a total of 99 models on representational alignment with the fMRI data in BMD.

**Measures of model-brain alignment.** The two most widely used approaches for measuring brain-model alignment are voxel-wise encoding (Güçlü & Van Gerven, 2015) and Representational Similarity Analysis (RSA) (Kriegeskorte et al., 2008). The former involves training a linear regression to predict voxel space from a model's feature space and conduct all analyses there, modeling each voxel response independently rather than how voxels respond together. The latter projects both spaces into a third common space of pairwise condition patterns; it works by computing a distance metric between the representations of each pair of test conditions for both voxel and feature space, forming the Representational Dissimilarity Matrix (RDM), which reflects differences between patterns of activity within populations of voxels. There are several influential works assessing model-brain alignment for static images, using both voxel-wise encoding (Eickenberg et al., 2017; St-Yves & Naselaris, 2018; La Tour et al., 2022) and RSA (Cichy et al., 2016; Dobs et al., 2019; King et al., 2019; Bartnik et al., 2024) as well as methods combining the two approaches (Dwivedi et al., 2021; Konkle & Alvarez, 2022; Conwell et al., 2022; 2024). Many of these works validate the image models' alignment to the brain's early-to-late processing hierarchy, as has also been shown with speech models (Millet et al., 2022), but it has not been seen if a similar mapping can be found in video models. *In this work,* we use RSA to derive the representational alignment between the video features of each neural network's layers and the voxels in each brain Region of Interest (ROI) in BMD, contrasting this alignment between early and late stages of visual processing.

**Model-brain alignment benchmarking.** Extensive benchmarking in the image domain was conducted by Conwell et al. (2022; 2024) on NSD, who decoupled multiple factors that give rise to differences between image models, specifically model architecture, diet (training dataset), and training task. They varied these one at a time while keeping all others constant, to assess which factors most influence model-brain alignment. A similar approach was followed in Wang et al. (2023a) to examine the effect of contrastive learning on model-brain alignment. There are also works extensively benchmarking neural networks on alignment to human behavior, using behavioral similarity judgment data (Muttenthaler et al., 2023; Marjieh et al., 2023), where the latter also includes three video action recognition models. In Mineault et al. (2021), a limited testing of five video action recognition models was performed to compare against their own model trained with a custom objective. More recently, a study by Garcia et al. (2024) focused on social-action videos and conducted a first, yet limited, video benchmarking by testing eight video models against 200 image models. *In this work,* we include 47 video models for action class recognition and 41+11 image models for object and action class recognition. Inspired by Conwell et al. (2022), we are interested in disentangling the role of different factors, and thus we vary the models' architecture by contrasting CNNs to Transformers, and the models' training dataset by contrasting three different datasets. Importantly, we control for the change in task when comparing object recognition image models to action recognition video models, through varying one factor at a time by further comparing with image models trained for action class recognition. In addition to Conwell et al. (2022), apart from involving dynamic stimuli, our analyses also address specific brain regions instead of the visual cortex as a whole. In contrast to Garcia et al. (2024), we compare an equally large number of video models to image models, on top of the important control of the training task and the comparison between different architectures and training dataset, and use RSA instead of voxel-wise encoding.

## 3 METHODOLOGY

In Figure 1 we show an overview of our methodology for measuring alignment of video models to the human brain.[1] In the next three sections, we describe the design choices of the alignment measure, the video models, and the brain imaging dataset.

### 3.1 ALIGNMENT BY REPRESENTATIONAL SIMILARITY ANALYSIS

We start by motivating our choice of metric to measure the alignment between a neural network and a brain region. To observe patterns of activity that consider the interactions between groups of voxels (multivariate analysis), either Representational Similarity Analysis (RSA) (Kriegeskorte et al., 2008) or a combined approach that performs RSA on top of voxel-wise encoding (mixedRSA or veRSA (Khaligh-Razavi et al., 2017)) are good candidates. Conwell et al. (2022; 2024) extensively compared these methods and showed that veRSA often fails to uncover differences between models in terms of brain alignment. This is likely due to the fact that voxel-wise encoding allows for re-weighting of the model activations while mapping them to the brain, thereby optimizing brain predictivity but obscuring how the model representations correlate with brain responses out of the box. Based on our intent to benchmark the emergent alignment of AI models, we choose RSA [2] as a stricter metric that can uncover more potential differences between models.

**Representational Dissimilarity Matrix (RDM) computation.** For a brain Region of Interest (ROI) of a human subject $s$ with voxel vectors $v_s$ and a model layer $l$ with feature vectors $f_l$, the brain RDM ($B_s$) and model RDM ($M_l$) are calculated as follows:

$$B_s^{ij} = 1 - r(v_s^i, v_s^j), \quad M_l^{ij} = 1 - r(f_l^i, f_l^j), \quad \forall i, j(i < j), \ 0 \le j < N, \tag{1}$$

where $N$ is the total number of videos and $r$ is the Pearson correlation, resulting in RDMs that are symmetric with size $N(N-1)/2$. To obtain $v_s$ we first average the voxels across measurements from all $K$ repetitions of a stimulus video $i$, so $v_s^i = \frac{1}{K} \sum_{k=0}^{K-1} v_{s,k}^i$. To create $f_l$ we first reduce the dimensionality of the original features to 100 Principal Components using Principal Component Analysis (PCA). We provide a comparison with Sparse Random Projection (SRP) and the full dimensions in Figure 16, Appendix D. No standardization is performed on the voxel and feature vectors, neither across videos nor across features to avoid distortions (Walther et al., 2016).

**Correlation of RDMs.** To derive the alignment of a model layer and a brain ROI, we correlate each layer RDM of each model ($M_l$) with each subject RDM of each ROI ($B_s$) using Spearman correlation ($\rho$), and then average the correlations across subjects for each model layer:

$$R_l = \frac{1}{M} \sum_{s=0}^{M-1} \rho(B_s, M_l), \tag{2}$$

where $M$ is the total number of subjects. In analyses where we show only one correlation value for the alignment of a whole model and a brain ROI, this is computed by $R = max_l(R_l)$, i.e., the highest-correlating layer in the model.

**Noise ceiling computation.** Because of individual subject variability in brain data, noise ceilings for each ROI are computed to compare model RSA scores against the maximum obtainable score given the inter-subject variability (Nili et al., 2014). For the lower noise ceiling (LNC) we compute a mean RDM across all subjects except one. Then we take the Spearman correlation of the left-out subject RDM and mean RDM, repeating for all the subjects and calculating the average. For the upper noise ceiling (UNC) we take the mean of all RDMs without removing subjects, compute the Spearman correlation of each subject RDM with the mean RDM, and average.

$$LNC = \frac{1}{M} \sum_{j=0}^{M-1} \rho(\frac{1}{M} \sum_{i=0, i \ne j}^{M-1} B_i, \ B_j), \ \ UNC = \frac{1}{M} \sum_{j=0}^{M-1} \rho(\frac{1}{M} \sum_{i=0}^{M-1} B_i, \ B_j) \tag{3}$$

The upper noise ceiling signifies perfect correlation for the amount of noise in this ROI's data, often referred to as the maximum amount of variance that can be explained.

---

[1] Due to licensing restrictions of the stimuli used in the brain dataset, the video frames shown in the figure are sourced from representative videos captured by the authors themselves and are not subject to copyright.

[2] We utilize the RSA implementation from the Net2Brain python library (Bersch et al., 2022).

Table 1: Model families benchmarked. We sample the same number of CNN and Transformer image object recognition models as in our maximal set of video models. We also test 10 image models trained on action recognition; this division is shown with the grouping on the right. Action recognition models are on Kinetics 400; those also available on other datasets are marked by $a,b$.

| **Image Object Recognition** | | | | **Action Recognition** | | | | |
|---|---|---|---|---|---|---|---|---|
| | **CNNs** | | **Transformers** | | **CNNs** | | **Transformers** | |
| 1 | AlexNet | 2 | CAiT | 6 | CSN | 2 | MViTv2$^b$ | |
| 2 | DenseNet | 2 | ConViT | 5 | I3D | 2 | TimeSformer | |
| 2 | EfficientNet | 2 | DEiT | 1 | R2P1D | 2 | Uniformer | |
| 2 | RegNet | 2 | MViTv2 | 2 | SlowFast | 2 | Uniformerv2$^a$ | Video |
| 4 | ResNet | 3 | Swin | 4 | Slow$^a$ | 1 | VideoMAE | |
| 2 | ResNeXt | 1 | Twins | 1 | TaNet | 2 | VideoMAEv2 | |
| 4 | VGG | 2 | ViT | 1 | TPN | 3 | VideoSwin$^a$ | |
| 2 | WideResNet | | | 5 | TSM$^b$ | | | |
| 2 | Inception | | | 2 | X3D | | | |
| 2 | RepVGG | | | | | | | |
| 2 | SeResNe(X)t | | | 4 | C2D | 1 | TimeSformer | Image |
| 2 | Xception | | | 4 | TSN$^b$ | 1 | TSN | |
| 27 | | 14 | | 27+8 | | 14+2 | | |

$^a$Availability also on Kinetics 710 (Carreira et al., 2019)
$^b$Availability also on Something-Something-v2 (Goyal et al., 2017)

**Statistical significance.** To test if model RDMs correlated significantly with brain RDMs, we performed permutation tests (Nili et al., 2014). For each model we permute the rows of all layer RDMs 1000 times using the same 1000 random permutations for all models and layers. We then calculate a null distribution by computing the Spearman correlations of all permuted RDMs with each subject RDM and average the resulting null distributions across subjects, separately for each model layer. For significance of a group of models against zero, we perform a two-tailed sign test between the median null distribution of all models in the group and the median observed Spearman correlation. To test for significant differences between two groups of models, we perform a two-tailed sign test between the null distribution created from the across-group differences in the within-group median distributions, and the observed difference in the medians of the two model groups' Spearman correlations. We correct for multiple comparisons between model groups by applying Bonferroni correction equal to the number of group pairwise combinations.

## 3.2 VIDEO MODELS

Our goal was to benchmark as many publicly released video models as possible, sampling from different architectures (e.g. CNNs, Transformers) as equally as possible, while differentiating the effects of optimizing for the action classification task and temporal modeling on brain alignment.

**Model choice.** In total, we benchmark 99 models; of these, 41 are image models trained for object recognition on ImageNet, 10 are image models trained for action recognition on Kinetics 400 (Kay et al., 2017), and 41 are video models trained for action recognition on Kinetics 400. The distinction between the last two is made based on whether the models treat time in a non-trivial way, where trivial is considered anything that makes completely separate computations per frame and only averages frame features before classification, or aggregates frames with static pooling operations at different stages. The remaining 7 models are trained for action recognition on other datasets, namely Kinetics 710 (Carreira et al., 2019) and Something-Something-v2 (Goyal et al., 2017). Image models trained on object recognition were ported from torchvision[3] and timm[4], while image and video models trained on action recognition were ported from mmaction2[5]. The main families of models used are listed in Table 1, while a full list of all the model versions used can be found in Appendix B.

**Model feature extraction.** We perform preprocessing according to the functions provided by the models' sources (torchvision, timm, or mmaction2). Regarding the temporal dimension, for image models trained on ImageNet we perform inference for all frames and then average the features, while action recognition image models were trained on sequences of frames as input samples (only

---

[3]https://pytorch.org/vision/main/models.html
[4]https://huggingface.co/models?library=timm
[5]https://mmaction2.readthedocs.io/en/latest/model_zoo/recognition.html

to aggregate with trivial pooling operations), so for those we perform inference on video inputs, as we do for video models. Because each model expects a specific length of timepoints sampled at a specific rate, different size batches of sub-clips are created per video and model, and the resulting sub-clip features are averaged for the whole video. We extract features from all higher-level blocks in the models (e.g., in a ResNet-type model of five blocks with four layers each, we extract features at the end of each block) and also include the final fully connected classification layer. We flatten the features after extraction, producing a single one-dimensional feature vector per layer.

## 3.3 BRAIN DATASET

We use the Bold Moments Dataset (BMD) (Lahner et al., 2024) consisting of whole-brain 3T fMRI recordings ($2.5 \times 2.5 \times 2.5$ mm voxels, resampled TR of 1s) from 10 subjects watching 1102 3s videos from the Moments in Time (Monfort et al., 2019) and Multi-Moments in Time (Monfort et al., 2021) video datasets. Extensive details on the fMRI data acquisition can be found in Lahner et al. (2024); a summary is provided in Appendix A. For each subject, 1000 videos were shown for 3 repetitions and those recordings make up the "training set", whereas 102 videos were shown for 10 repetitions and make up the "test set". In our analysis, we only use the 102 videos of the test set, whose high number of repetitions allows for the application of RSA. We report partial results on the 1000 video training set in Figure 17, Appendix D, showing noise increase from insufficient repetitions. The test set videos are sensibly representative of the videos present in the whole dataset. Specifically, roughly 25% of the videos are actions of children and 22% actions of animals, another 22% people doing sports, 10% are scenes with motion but no visible humans or animals (e.g. a waterfall, a car), 6% people cooking, and 6% people performing some other manual labor.

**Preprocessing and Regions of Interest (ROIs).** We used preprocessed data provided by (Lahner et al., 2024). A summary of the most important preprocessing steps, including anatomical alignment and ROI definitions, is again provided in Appendix A. From the available brain ROIs, we first joined the two hemispheres by concatenating the voxels corresponding to each pair of areas, and then also joined the dorsal and ventral parts of V1 and V2. We group ROIs based on their anatomical location on the brain (Grill-Spector & Malach, 2004), in four groups: Early Visual Cortex (includes areas V1, V2, V3v, V3d), Ventral-Occipital Stream (hV4, OFA, LOC, FFA, PPA), Dorsal Stream (V3ab, IPS0, IPS1-3, RSC), and Lateral Stream (EBA, TOS, MT, STS). Areas in Early Visual Cortex are considered early areas of processing, hV4 and V3ab are considered intermediate, and the rest are considered late areas of visual processing, based on the flow of information through the brain.

## 4 RESULTS

Differences in brain alignment between 99 neural networks processing videos could be related to a number of factors. In the following sections we systematically examine the following factors: temporal modeling, classification task, architecture design, and training dataset. Finally, we also investigate how brain alignment relates to the models' computational complexity.

**Video vs. Image models, controlling for the classification task.** First, we compare models by varying the temporal modeling (Video vs. Image) and controlling for the classification task (object vs. action recognition) while keeping the training dataset constant (action recognition on Kinetics 400). In Figure 2a we observe that in the Early Visual Cortex, video models score significantly higher than the image models, while image models trained on action recognition may even capture less variance than the image models trained on object recognition. The control for image models trained on action recognition decouples the effect of classification objective from temporal modeling, and shows that the latter is the determining factor out of the two for RSA scoring in the early visual cortex. In later areas it can be seen that here the classification objective exerts more influence on the RSA score, as the video models do not fare much better than image models trained on action recognition, and those in general score higher than image models trained on object recognition. Out of all the later area groups, the Lateral stream shows the least significant differences. In relation to the noise ceilings, we observe that models are able to explain the largest amount of the total explainable variance in the Early Visual Cortex out of all brain regions (see Figure 13, Appendix D for a re-scaled plot by the UNC). In Figure 2b we notice that all three model categories exhibit an early-to-late hierarchy, with more shallow layers correlating the highest with early brain regions, and deeper layers correlating the highest with late brain regions. The mid-network representations

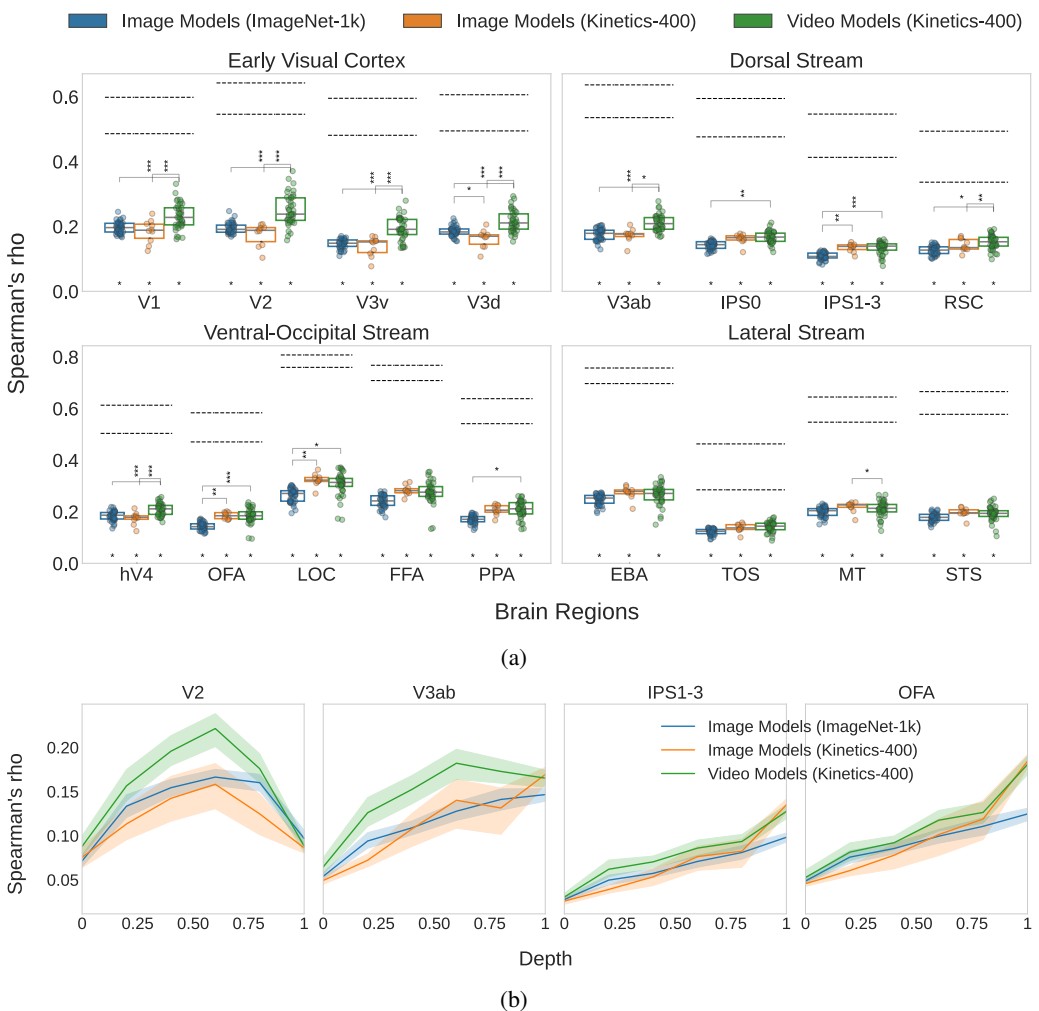

Figure 2: **Video vs. Image models, controlling for the classification task:** We find that temporal modeling is the important factor for representational alignment to early brain regions, while action space optimization for alignment to late brain regions, evaluating the RSA score for a total of 92 models. In 2a we choose the best scoring layer for each model and show all 17 brain regions. Stars at $y = 0$ and on top of brackets indicate statistical significance against zero and significance of pairwise differences respectively ($* : p < 0.05$, $** : p < 0.01$, $*** : p < 0.001$). Horizontal dashed lines show the upper and lower noise ceilings for each region. In 2b we show different model layer depths for regions chosen by zooming in on significant differences from 2a and sampling region groups evenly, the rest reported in Appendix C.

learned by the video models correlate better with V2 than those of image models, regardless if the latter are trained for action recognition. On the other hand for OFA, it is the representations in the classification layer of action recognition models that manage to match the brain better, even if there is no temporal modeling involved. In Figure 3 we point out that the top models are MViT, CSN, and SlowFast in V2, and CSN, TimeSformer, and VideoSwin in LOC. In V2 and for a portion of the subjects, the model MViT_v2_S comes close to capturing almost all of the explainable variance.

**Varying the architecture.** Next, we compare models by varying the architecture design (CNNs vs. Transformers), keeping temporal modeling, classification task, and training dataset all constant (video action recognition models trained on Kinetics 400). In Figure 4 we see that, in terms of the maximum score, Transformers and CNNs appear to be mostly equivalent, either of the two architectures taking the lead in different ROIs. Nevertheless, an interesting pattern appears when we analyze the changes in correlation across the models' layers. In V2 Transformers compute high correlating representations at a very shallow depth in the network (around 0.1 of the total depth), while CNNs much later, at around 0.6 of the total depth. In EBA, CNNs show a much clearer

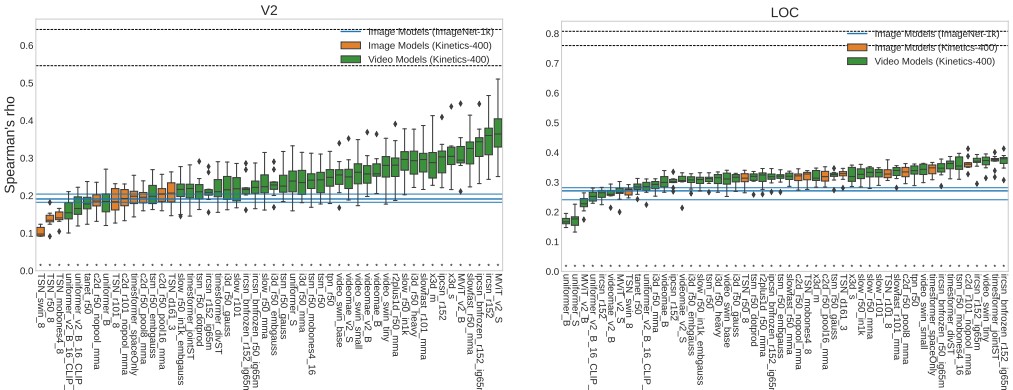

Figure 3: Ranking of all the Kinetics-400 action recognition models in RSA scoring order, against the baseline of image object recognition models. Top models in V2 are MViT, CSN, and SlowFast, while in LOC they are CSN, TimeSformer, and VideoSwin. Error bars signify variation across subjects. One early and one late region are shown (the two highest overall), the rest in Appendix C.

hierarchy, as correlation gradually increases with model depth, while Transformers have relatively stable representations up until the classification layer, where the score increases more abruptly. Still in EBA, Transformers explain more variance than CNNs when comparing only very shallow layers.

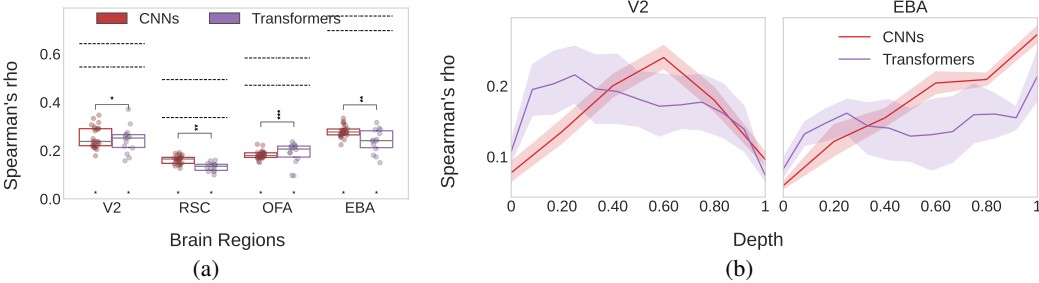

Figure 4: **Varying the architecture:** CNNs and Transformers are overall equivalent but exhibit striking differences in their score distribution across layer depths. We show RSA scores of 27 CNNs and 14 Transformers, for the best scoring model layer (4a) and all layer depths (4b). Brain regions shown are those exhibiting the most significant differences, sampling evenly from brain region groups. Exhaustive results can be found in Appendix C.

**Varying the training dataset.** Here, we compare models by varying the training dataset (Kinetics 400 vs. Kinetics 710 and Kinetics 400 vs. Something-Something-v2) and keeping all other factors constant through only including the exact same models trained on the two different datasets in each pair of dataset comparisons. In Figure 5a we first observe that using models trained on an extended and enhanced version of the same dataset such as Kinetics 710 in relation to Kinetics 400, largely has no effect on brain alignment. However, when testing models trained on a more different domain, such as Something-Something-v2, some interesting differences emerge. In particular, there is a significant advantage of Kinetics 400 against Something-Something-v2 in FFA and OFA, and a somewhat similar pattern in LOC. In fact, this seems to show up both in shallow and deep layers of the models (Figure 5b). Considering that Something-Something-v2 is a dataset that never shows faces and that FFA is the functionally selective region for faces, this result identifies that a dataset bias from the models' training can indeed be transferred to model-brain alignment for an ROI that is functionally selective specifically for this bias.

**Relation to computational complexity.** Last, we investigate how the brain correlation of action recognition models relates to their computational complexity (in FLOPs). In Figure 6, we report a moderate but consistent negative correlation of model FLOPs to model-brain alignment especially in late ROIs of the Lateral and Dorsal Streams, with significant negative correlations observed in six ROIs, the top four of which are shown in 6b. In 6a we observe that although the significance of the negative relation is not present in all ROIs, the relation itself is mostly consistent throughout the brain regions. A similar investigation for model parameters and model accuracy was performed but no consistent significant correlations were found (see Figure 21, Appendix E).

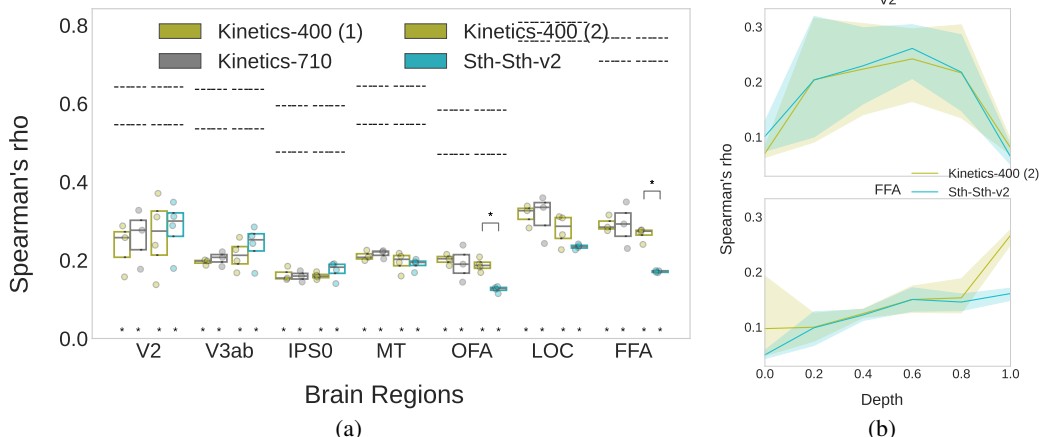

(a)

(b)

Figure 5: **Varying the training dataset:** Training dataset biases transfer to alignment with functionally selective brain areas. We compare the same three models trained on Kinetics 400 and Kinetics 710, and the same four models trained on Kinetics 400 and Something-Something-v2. Models trained on Kinetics 400 are numbered as (1) and (2) to indicate two different sets of models compared with Kinetics 710 and Something-Something-v2 respectively, and the comparisons are also separated with a gap. In 5a the highest-scoring layers are shown, selecting a representative sample of regions (the rest in Appendix C), and in 5b we zoom in on two regions to show all layer depths.

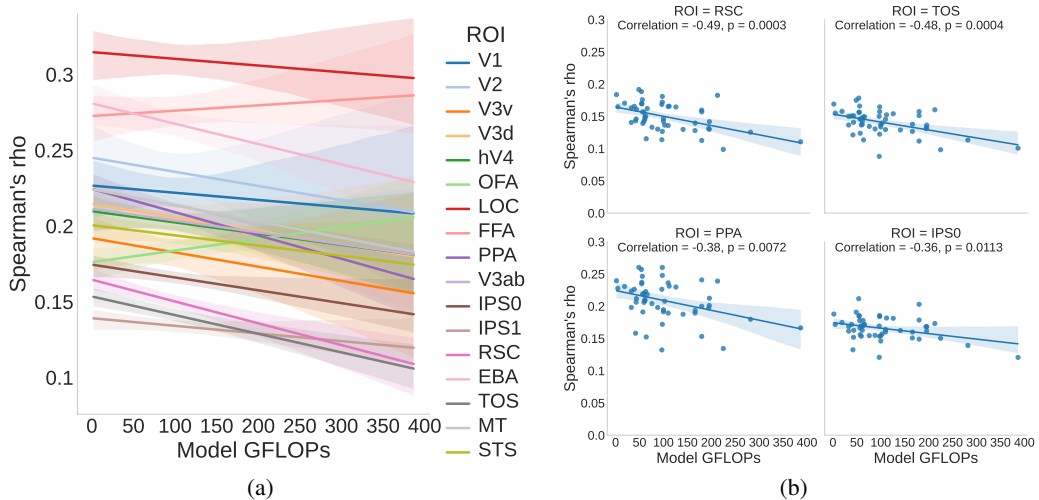

(a)

(b)

Figure 6: **Relation to computational complexity:** We show a significant negative correlation of model FLOPs to alignment in several high-level brain areas. We correlate the RSA scores of video models to model FLOPs, using the best RSA scoring layer for each model and report all brain regions (6a), as well as highlight those with the most significant correlations (6b).

## 5 DISCUSSION

**Decoupling the role of temporal modeling and action space optimization.** For alignment of video models to brains watching videos, and for our sample of models, brains, and videos, different factors are important at different stages of visual processing. In early areas, temporal modeling improves alignment, while optimizing for action recognition does not. The latter on its own can even result in slightly worse alignment than object recognition models, hinting to a potential degradation of low-level features during action optimization (image action recognition models are pretrained on ImageNet first). The benefit of temporal modeling for alignment with early brain areas is consistent with neuroscience literature suggesting that these areas are more sensitive to short time-scale change in the input, while late areas are more temporally invariant (Hasson et al., 2008; Groen et al., 2022; Brands et al., 2024). There, improvement over object recognition image models only comes from optimizing for action recognition and is only reflected in the models' classification layer. This could mean (1) that the information captured in late brain areas is dominated by semantics and particularly action categories (Lahner et al., 2024). Or (2) that current state-of-the-art video models

fail to capture the more fine-grained high level video representations present in the brain, aside from action labels. Future work is needed to resolve these competing accounts. In line with Mineault et al. (2021), we find no indication that action information affects alignment more for areas involved in navigation such as RSC or motion area MT. Garcia et al. (2024) observed a different pattern of results from ours, finding improved alignment for video models mainly in MT and EBA. This could reflect methodological differences (e.g. fMRI dataset, dimensionality reduction method, brain alignment metric). Our findings are consistent with a recent video reconstruction study (Lu et al., 2024a) which shows that intermediate and late regions contribute more to the decoding of video semantics, while early areas more to the decoding of low-level structure and motion in the videos.

**Comparing early and late representations of CNNs and Transformers.** CNNs exhibit a better hierarchy overall, as there is a clear mid-depth peak for early regions and gradual improvement as depth increases for late regions. Transformers however, achieve an impressive correlation to early regions even from one tenth of layer depth. This also holds for image CNNs and Transformers (Figure 19). Why is the top-aligned layer mid-depth in CNNs, and not earlier? It does not appear to be a matter of network size or depth (see Figures 20, 22) - therefore, an interesting hypothesis is that this relates to the dynamic nature of the stimuli, as in image fMRI studies it was not observed (Nonaka et al., 2021). This hypothesis requires further validation on (multiple) other video fMRI datasets. Conversely, why are Transformers more aligned earlier in the network than CNNs? Raghu et al. (2021) showed that representations in layers 0-60 of a CNN are more similar to layers 0-30 of a Transformer, and CNN layers 60-120 more similar to Transformer layers 30-140. Transformers also exhibited more stable representations across layers, while CNNs showed a clear divide between early and late layers. In Tuli et al. (2021), more human-like error patterns of Transformers indicated a higher behavioral alignment - we find that alignment in high-level areas based on semantic representations is mostly equivalent across both architectures. In future work it is also worth investigating hybrid models (e.g. Transformers using both convolution and self-attention), as there are indications these might achieve a balance between the two architectures (Figure 15).

**Relating training biases and computational efficiency to alignment.** Our comparison of brain alignment between models trained on different action datasets shows that dataset biases related to a certain functional selectivity can be transferred in brain alignment with the respective functionally selective brain area. This highlights the importance of choosing models trained on a dataset that is representative (1) of the videos humans watch in the experiment, and (2) of the hypotheses tested when measuring alignment. Our observation that computationally expensive models are less aligned to higher brain areas leads us to conclude that human-like semantics are more achievable with computationally efficient models. This is important for machine learning research that aims to build increasingly efficient models for increasingly complex tasks, as it is an indication that more computational resources may not be needed to compute human-like high-level representations. Ideas in this direction include using the top-aligned models as starting points for further component ablations towards alignment and efficiency, as well as pruning models to preserve alignment.

**Limitations and future work.** We base our choice of alignment metric on the comparison of RSA and veRSA made in Conwell et al. (2022), but this was on static images and it could be worthwhile to make the same comparison for our video data, as well as comparison between other metrics. RSA does not examine possible gains in predictivity as a result of linear feature re-weighting. Next, we benchmark video models that are part of a library and not all publicly available video models, which limits the comparisons we could make; to show a controlled comparison of learning paradigms we would need multiples of the same architecture in supervised, contrastive, and self-supervised variants. Our results are based on a single fMRI dataset, and not validated across multiple fMRI experimental setups. BOLD signals are also indirect measurements of brain activity, which in turn is an indirect measurement of the brain's representations - these are latent variables that cannot be measured. Our fMRI analyses were conducted in volume space; surface-based analysis could potentially provide better predictions of brain function (Glasser et al., 2016; Coalson et al., 2018). Importantly, fMRI lacks good temporal resolution which is crucial to investigate temporal modeling at finer detail. Future steps in that direction would be to make use of all temporal samples available in the fMRI data (3 TRs) and collection of EEG/MEG data for the same videos, to perform alignment benchmarking jointly on fMRI and EEG/MEG - our work sets the foundations for such future studies by already uncovering differences in models performing temporal modeling with fMRI. Given that motion and imagery cues in videos likely also engage non-visual brain regions, another promising direction for future work is to study representational alignment in areas outside visual cortex.

## REPRODUCIBILITY STATEMENT

Open-source code to reproduce this benchmarking study can be found in the github repository `https://github.com/SergeantChris/hundred_models_brains`, along with detailed installation and usage instructions.

### ACKNOWLEDGMENTS

This work is supported by an ELLIS Amsterdam Unit grant to IIAG. CS[†] acknowledges travel support from the European Union's Horizon 2020 research and innovation program under ELISE Grant Agreement No 951847.

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

APPENDIX

A  BRAIN DATASET DETAILS

Below, we provide a summary of the Methods section of the BMD dataset (Lahner et al., 2024).

**Participants** Ten healthy volunteers (6 female, mean age ± SD = 27.01 ± 3.96 years, sex self-reported) with normal or corrected-to-normal vision participated in the experiment. All participants gave informed consent and were screened for MRI safety. The experiment was conducted in accordance with the Declaration of Helsinki and approved by the local ethics committee.

**Stimuli** The stimulus set consisted of 1102 videos in total and were sampled from the Memento10k dataset41. Each video was square-cropped and resized to 268×268 pixels. Videos had a duration of 3 s and frame rates ranging from 15 to 30 frames per second. Videos were manually selected from the Memento10k dataset by two human observers to encompass videos that contained movement (i.e., no static content), were filmed in a natural context, and represented a wide selection of possible events a human might witness. The 1102 videos selected for the main experiment were split into a training and a testing set; 102 videos were randomly chosen for the testing set. Subjects additionally viewed a separate set of colored, naturalistic videos (18 s length, composed of 6 3-second videos) corresponding to one of five categories (faces, bodies, scenes, objects, and scrambled objects) in order to functionally localize each subject's category selective regions of interest (ROIs, see below).

**MRI data acquisition** The MRI data were acquired with a 3 T Trio Siemens scanner using a 32-channel head coil. During the experimental runs, functional T2*-weighted gradient-echo echo-planar images (EPI) were collected (TR = 1750 ms, TE = 30 ms, flip angle = 71°, FOV read = 190 mm, FOV phase = 100%, bandwidth = 2268 Hz/Px, resolution = 2.5 × 2.5 × 2.5 mm, slice gap = 10%, slices = 54, multi-band acceleration factor = 2, ascending interleaved acquisition). Additionally, a structural T1-weighted image (TR = 1900 ms, TE = 2.52 ms, flip angle = 9°, FOV read = 256 mm, FOV phase = 100%, bandwidth = 170 Hz/px, resolution = 1.0 × 1.0 × 1.0 mm, slices = 176 sagittal slices, multi-slice mode = single shot, ascending) and T2-weighted image (TR = 7970 ms, TE = 120 ms, flip angle = 90°, FOV read = 256 mm, FOV phase = 100%, bandwidth = 362 Hz/Px, resolution = 1.0 × 1.0 × 1.1 mm, slice gap = 10%, slices = 128, multi-slice mode = interleaved, scending) were obtained as high-resolution anatomical references. Dual echo fieldmaps (TR = 636 ms, TE1 = 5.72 ms, TE2 = 8.18 ms, flip angle = 60°, FOV read = 190 mm, FOV phase = 100%, bandwidth = 260 Hz/Px, resolution = 2.5 × 2.5 × 2.5 mm, slice gap = 10%, slices = 54, ascending interleaved acquisition) were acquired at the beginning of every session to post-hoc correct for spatial distortion of functional scans induced by magnetic field inhomogeneities. Subjects completed a total of 5 separate fMRI sessions on separate days. Session 1 consisted of structural scans, functional localizer runs, and functional resting state scans all interspersed. Sessions 2–5 consisted of the main functional experimental runs where the subjects viewed the training and testing set videos.

**MRI data preprocessing** Raw MRI data was converted to BIDS format63 and preprocessed using the standardized fMRIPrep preprocessing pipeline (Esteban et al., 2019). Lahner et al. (2024) provide preprocessed data from two different pipelines, with MRI activations in Version A being calculated using Finite Impulse Response estimates, while Version B uses GLMsingle (Prince et al., 2022). We use Version B, as the authors recommend it for being higher quality. The full fMRIPrep-generated preprocessing report is found in Lahner et al. (2024). Briefly, it consisted of anatomical data preprocessing (intensity non-uniformity correction, skull-stripping, surface segmentation, and volume-based spatial registration to standard MNI152N-Lin2009cAsym space) and functional data preprocessing (B0-nonuniformity correction, head-motion estimation, slice-time correction, and registration to the anatomical reference, with all spatial transformations concatenated into a single step), generating preprocessed BOLD runs in standard MNI space. Functional localizer scans were spatially smoothed (9 mm FWHM) while the main experimental data remained unsmoothed.

**Brain response estimation** A General Linear Model (GLM) was used to estimate single-trial beta estimates for each video. First, the main experimental runs were temporally interpolated from their acquisition TR of 1.75 seconds to a TR of 1 second to time-lock volume sampling to stimulus presentations. The interpolated fMRI time series, stimulus onsets, and stimulus durations (modeled with 3s durations) for each session separately were input to GLMsingle (Prince et al., 2022), which estimates single-trial beta values by (1) fitting an optimal Hemodynamic Response Function (HRF) to each voxel from a library of HRFs, (2) identifying nuisance regressors from a noise pool that

maximally explain variance, and (3) implementing fractional ridge regression to improve estimates in a rapid event-related design. The resulting single-trial beta estimates were normalized within each scanning session using the session's training set mean and standard deviation.

**ROI definitions** A set of 23 ROIs (each separated by left and right hemispheres) previously known to be driven by dynamic stimuli spanning visual and parietal cortices were defined by Lahner et al. (2024) by creating a non-overlapping parcellation composed of parcels resampled from several anatomical atlases (Glasser et al., 2016; Wang et al., 2015; Julian et al., 2012), into standard MNI152NLin2009cAsym space. Then, brain activations estimated from the independent functional localizer runs were used to identify the top 50% of activated voxels within each of the corresponding parcels. This ROI definition method facilitates inter-subject modeling by ensuring all ROIs were defined for each subject and each ROI contained the same number of voxels across subjects. The defined ROIs were V1v, V1d, V2v, V2d, V3v, V3d, hV4, V3ab, IPS0, IPS1-3, BA2, 7AL, PFt, PFop, and MT, EBA, LOC, PPA, RSC, STS, OFA, FFA, and STS, separately for each hemisphere.

## B  MODEL DETAILS

In this section we explicitly list all the models benchmarked. In the main paper only the model families are reported, whereas here all the models in those families are shown, including different backbones, sizes, and configurations.

Table 2: Exhaustive account of all models benchmarked.

| Image Object Recognition | | Action Recognition | |
|---|---|---|---|
| **CNNs** | **Transformers** | **CNNs** | **Transformers** |
| AlexNet | CAiT_S | IR_CSN_R152 | MViTv2_S[b] |
| DenseNet161 | CAiT_XXS | IR_CSN_R152_BNfrozen_IG65M | MViTv2_B[b] |
| DenseNet201 | ConViT_S | IR_CSN_R50_BNfrozen_IG65M | TimeSformer_DivST |
| EfficientNetB3 | ConViT_B | IR_CSN_R152_IG65M | TimeSformer_JointST |
| EfficientNetB6 | DEiT_S | IP_CSN_R152_IG65M | Uniformer_S |
| RegNetX16gf | DEiT_B | IP_CSN_R152 | Uniformer_B |
| RegNetY8gf | MViTv2_S | I3D_R50 | Uniformerv2_B[a] |
| ResNet34 | MViTv2_B | I3D_R50_dotprod | Uniformerv2_B_k710pre |
| ResNet50 | Swin_T | I3D_R50_embgauss | VideoMAE_B |
| ResNet101 | Swin_S | I3D_R50_gauss | VideoMAEv2_S |
| ResNet152 | Swin_B | I3D_R50_heavy | VideoMAEv2_S |
| ResNeXt50 | Twins_pcpvt_B | R2P1D_R50 | VideoSwin_T |
| ResNeXt101 | ViT_S | SlowFast_R50 | VideoSwin_S[a] |
| VGG11 | ViT_B | SlowFast_R101 | VideoSwin_B |
| VGG11BN | | Slow_R50 | |
| VGG19 | | Slow_R101 | |
| VGG19BN | | Slow_R50_IN1k[a] | |
| WideResNet50 | | Slow_R50_IN1k_embgauss | |
| WideResNet101 | | TaNet_R50 | |
| InceptionV3 | | TPN_R50 | |
| InceptionV4 | | TSM_R50[b] | |
| RepVGGa2 | | TSM_R50_dotprod | |
| RepVGGb2 | | TSM_R50_embgauss | |
| SeResNet50 | | TSM_R50_gauss | |
| SeResNeXt50 | | TSM_MobOne_s4 | |
| Xception41 | | X3D_S | |
| Xception71 | | X3D_M | |
| | | | |
| | | C2D_R50_nopool | TimeSformer_SpaceOnly |
| | | C2D_R101_nopool | TSN_Swin |
| | | C2D_R50_pool8 | |
| | | C2D_R50_pool16 | |
| | | TSN_R50[b] | |
| | | TSN_R101 | |
| | | TSN_D161 | |
| | | TSN_MobOne_s4 | |

[a] Availability also on Kinetics 710 (Carreira et al., 2019)
[b] Availability also on Something-Something-v2 (Goyal et al., 2017)

# C  ALL REGIONS OF INTEREST

In this section we exhibit all Regions of Interest (ROIs) for analyses where a subset of the ROIs was shown in the main results.

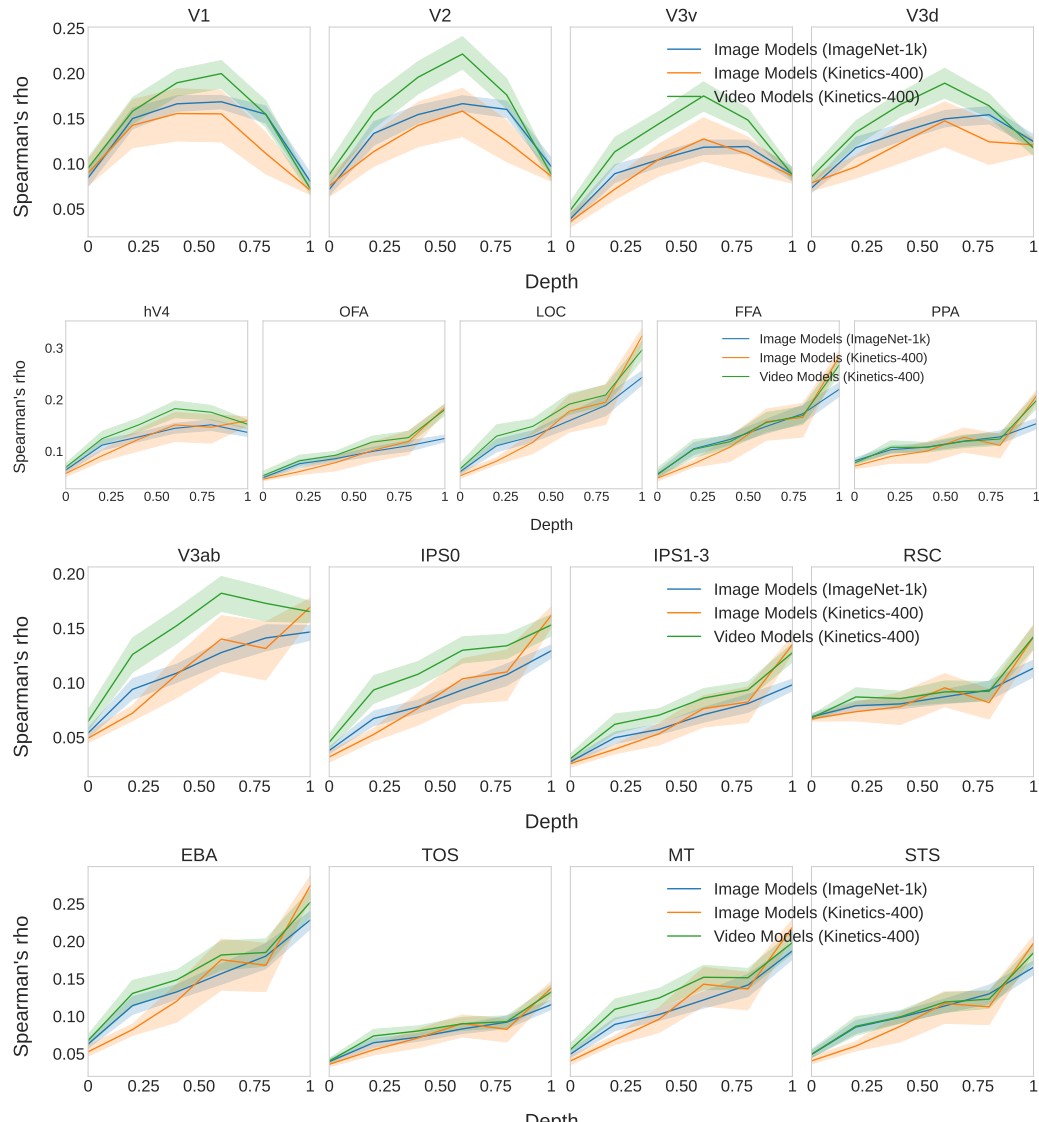

Figure 7: Expanding Figure 2b to show, for all ROIs, the comparison across layers of object recognition image models and action recognition image and video models.

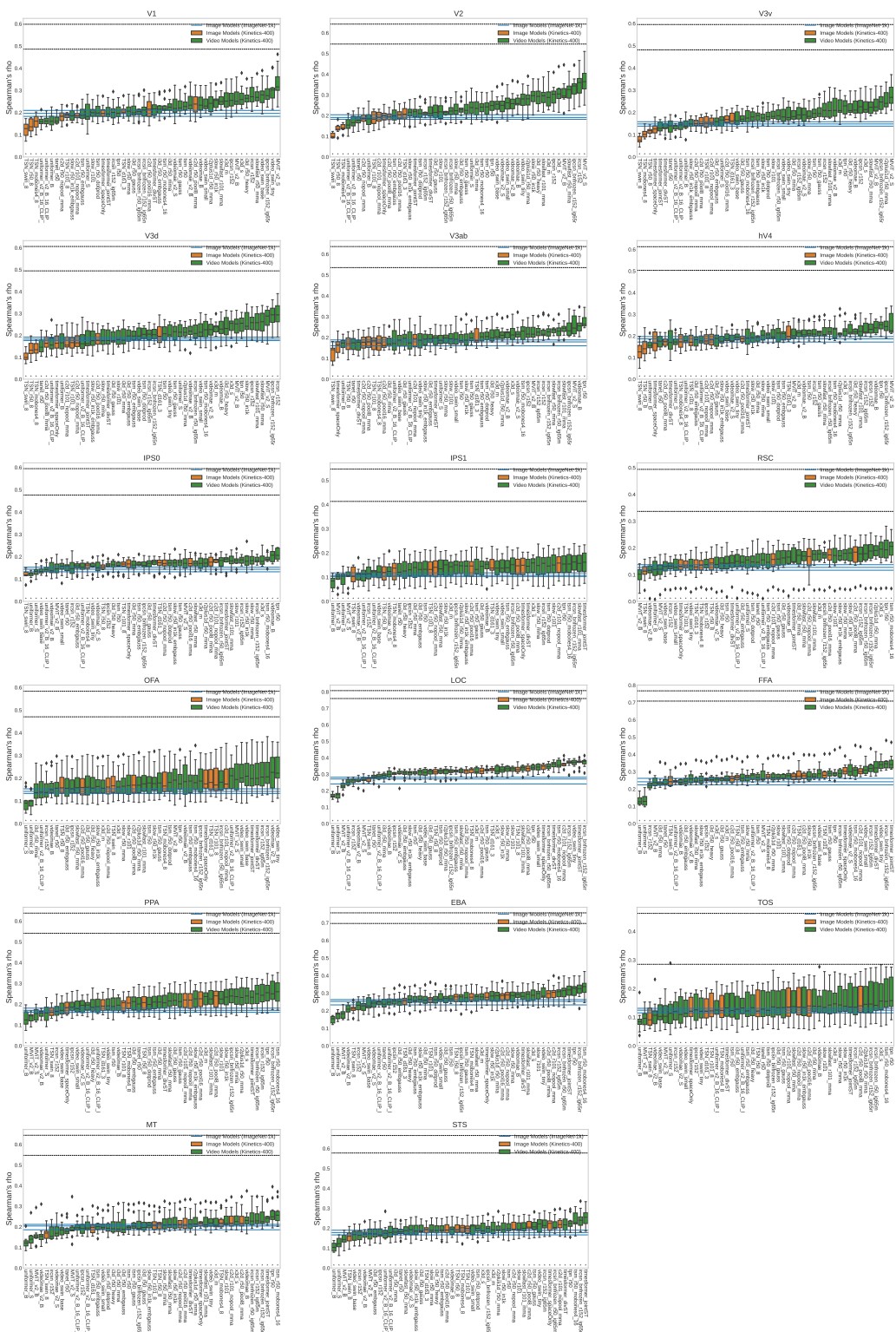

Figure 8: Expanding Figure 3 to show model ranking for all ROIs.

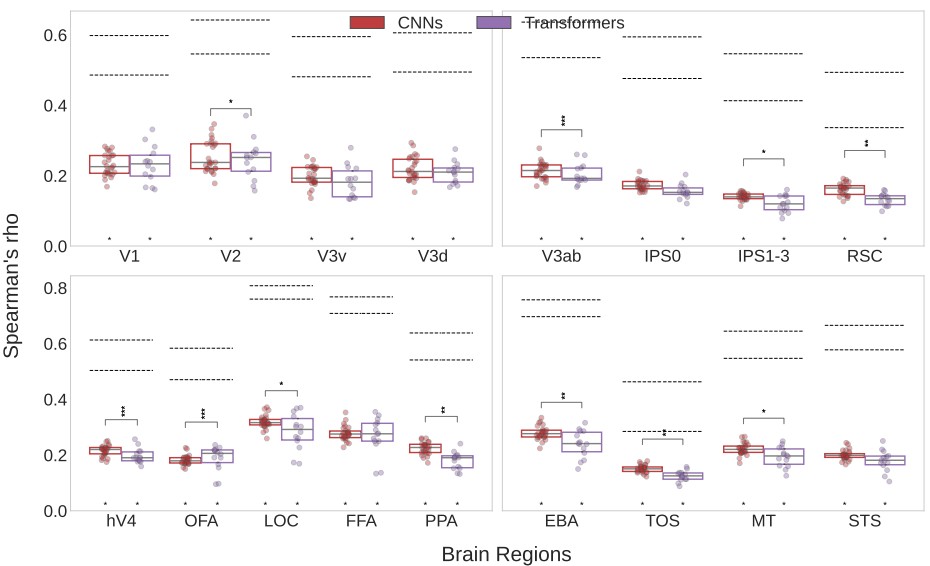

Figure 9: Expanding Figure 4a to show all ROIs for the comparison of CNNs and Transformers.

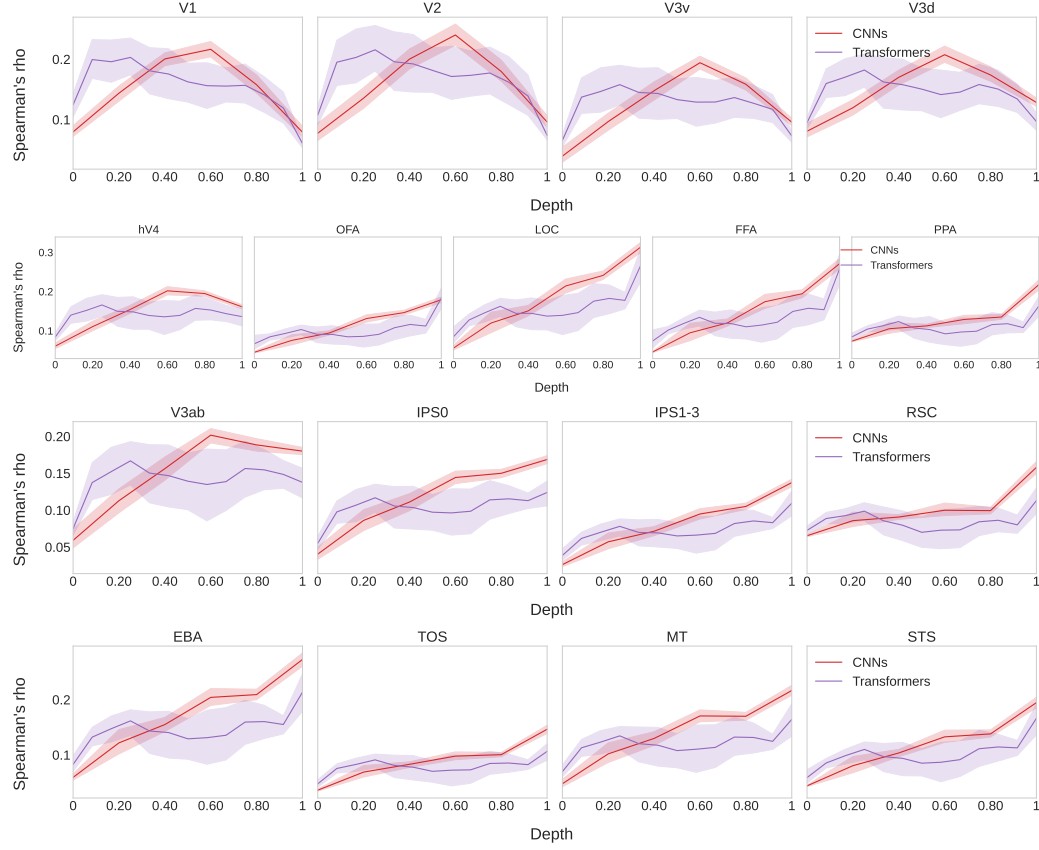

Figure 10: Expanding Figure 4b to show all ROIs for the comparison of CNNs and Transformers.

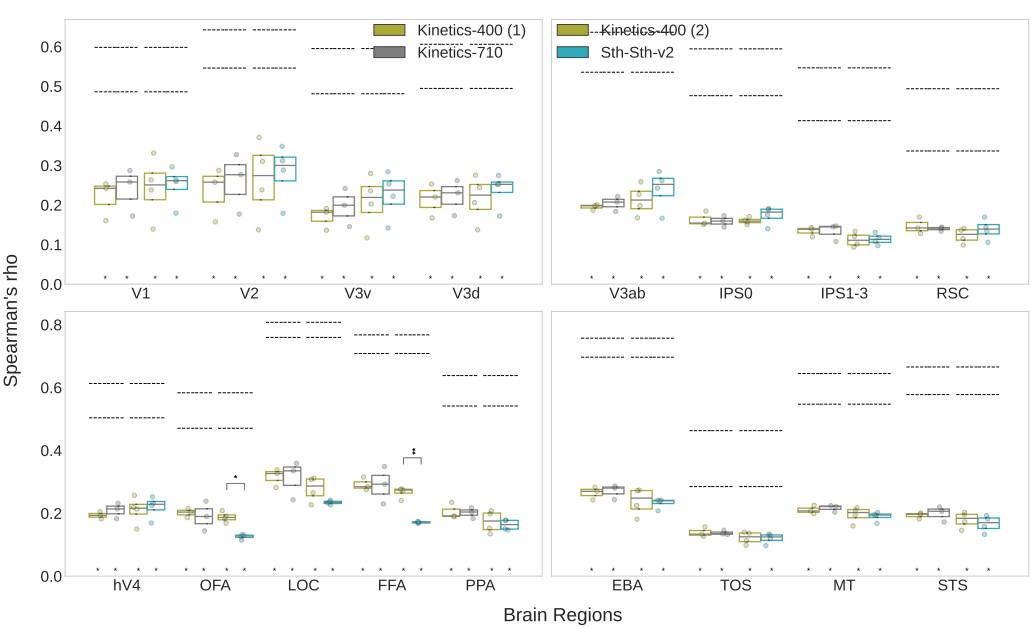

Figure 11: Expanding Figure 5a to show all ROIs for the comparison of training datasets.

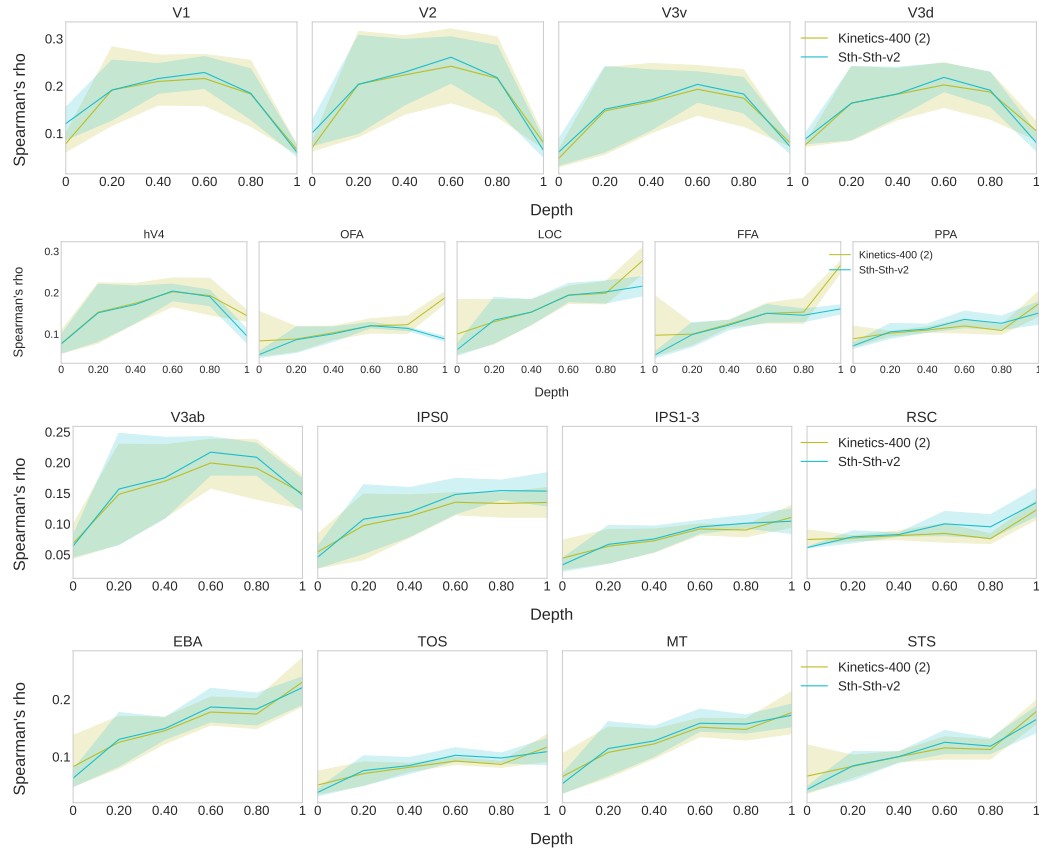

Figure 12: Expanding Figure 5b to show all ROIs for the layer-wise comparison of training datasets.

## D  ADDITIONAL VIEWS

In this section we include alternative ways to produce or view the results of our benchmarking, including different visualization, dimensionality reduction, or set of models.

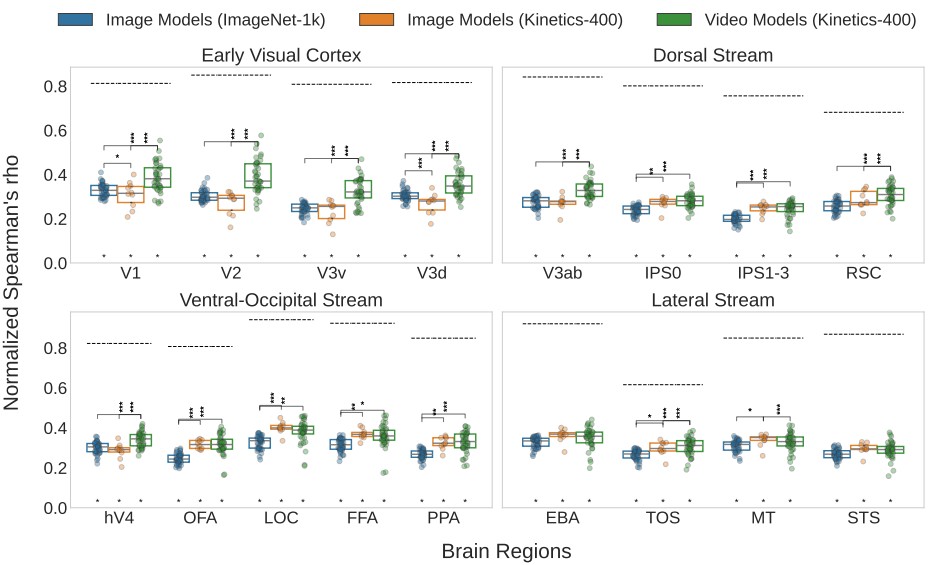

Figure 13: Version of Figure 2a, re-scaled to the Upper Noise Ceiling. With this visualization the amount of variance explained is more evident, but the information of ROI noise and absolute correlation is lost.

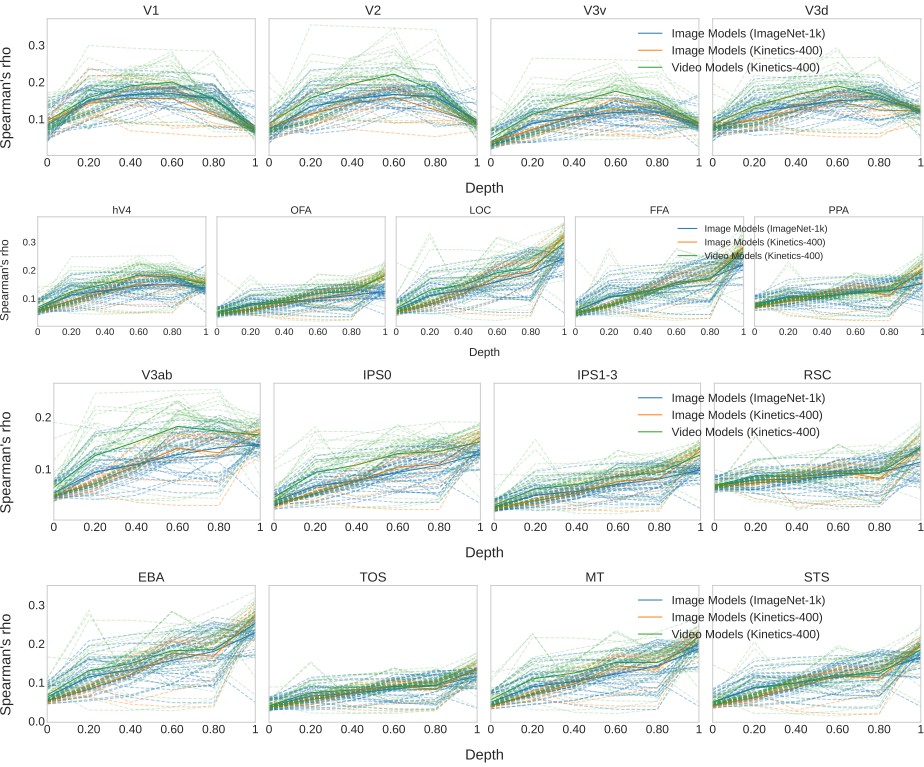

Figure 14: Version of Figure 7 with all individual models displayed as separate lines.

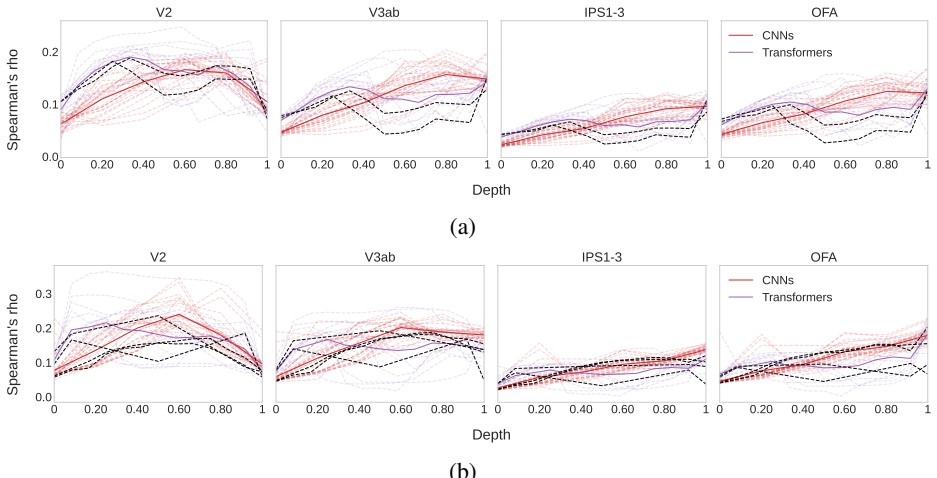

Figure 15: Comparing hybrid Transformers that include convolution. (a) shows image object recognition models, and black marks ConViT (d'Ascoli et al., 2021). (b) shows video action recognition models, and black marks Uniformer (Li et al., 2023b). ConViT's alignment is similar to the other image Transformers, while Uniformer seems to fall in between video CNNs and Transformers.

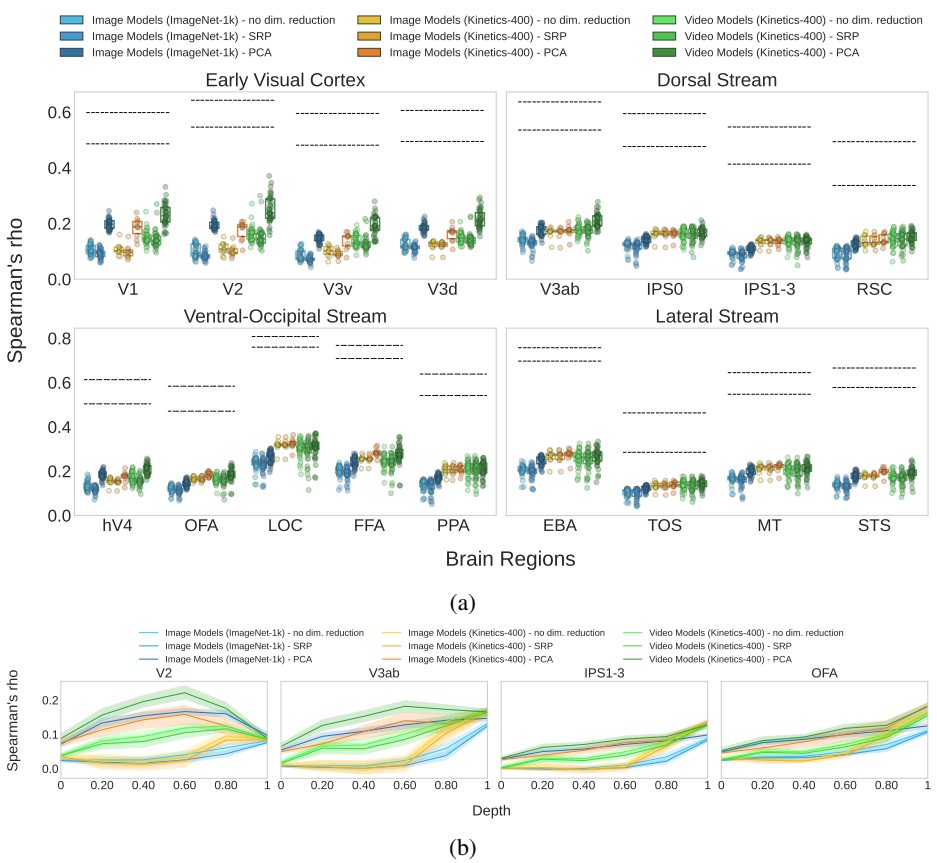

Figure 16: Dimensionality reduction method comparison on the analysis from Figure 2; using Principal Component Analysis (PCA) vs. using Sparse Random Projection (SRP) vs. no dimensionality reduction. Overall, models' RDMs match the brain less when constructed with full dimensions than when keeping important components - same with SRP where the dimensionality is still high.

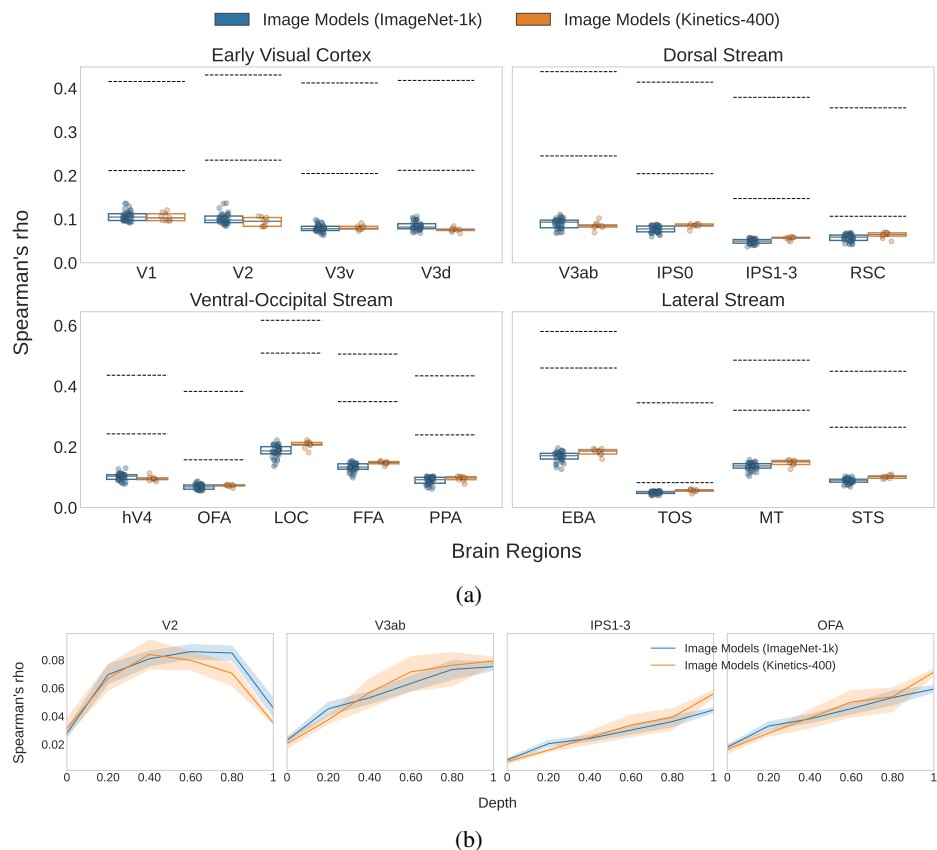

Figure 17: Image model results (analysis from Figure 2) with the 3-repetition 1000-video training set instead of the 10-repetition 102-video test set. Both noise ceilings and model scores drop significantly from Figure 2 due to increased measurement noise from the fewer stimulus repetitions.

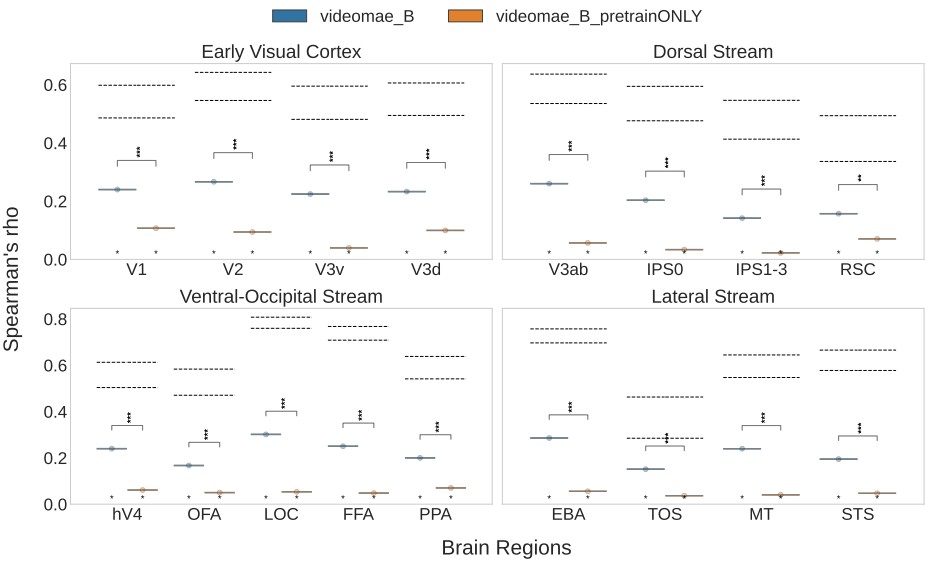

Figure 18: Comparison of the self-supervised model videomae_B, with and without (videomae_B_pretrainONLY) supervised finetuning on Kinetics 400. Here, it seems that supervised finetuning is crucial for alignment - however, a general conclusion should not be drawn from this one model comparison and future work is needed to investigate this further.

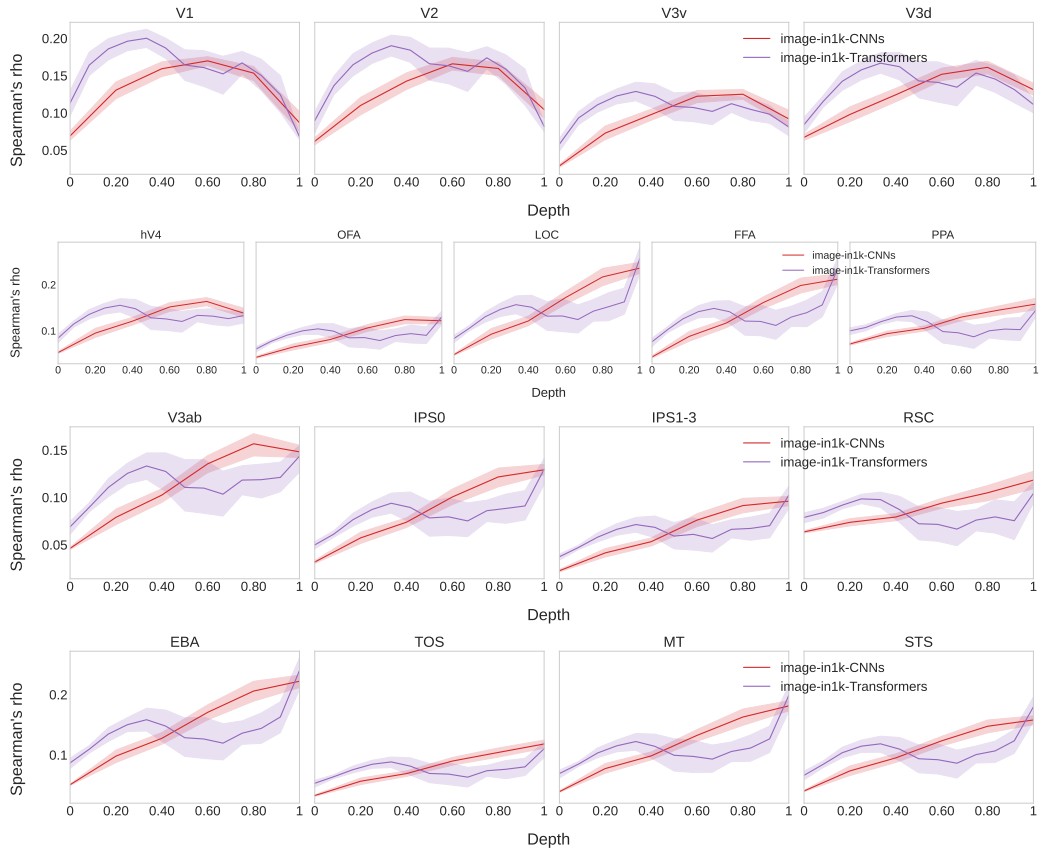

Figure 19: Layer hierarchy for image object recognition CNNs and transformers, plotted as in Figure 4b. Similar to the pattern displayed by video action recognition models, here transformers also exhibit an alignment peak to Early Visual Cortex much earlier in network depth than CNNs.

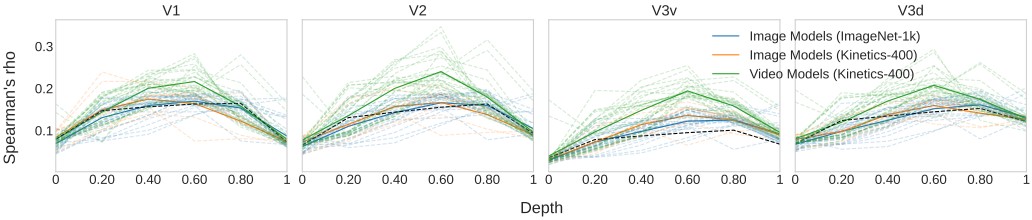

Figure 20: Only showing CNNs, plotted in separate lines. AlexNet, marked in black, also does not exhibit an early-layer peak in Early Visual Cortex regions. The only CNNs having an early-layer peak in V1 are C2D and TPN, which both have a ResNet50 backbone and so are not particularly shallow.

# E  RELATION TO PARAMETERS & ACCURACY

In this section we provide results for the relation of the RSA score to model parameters and model accuracy.

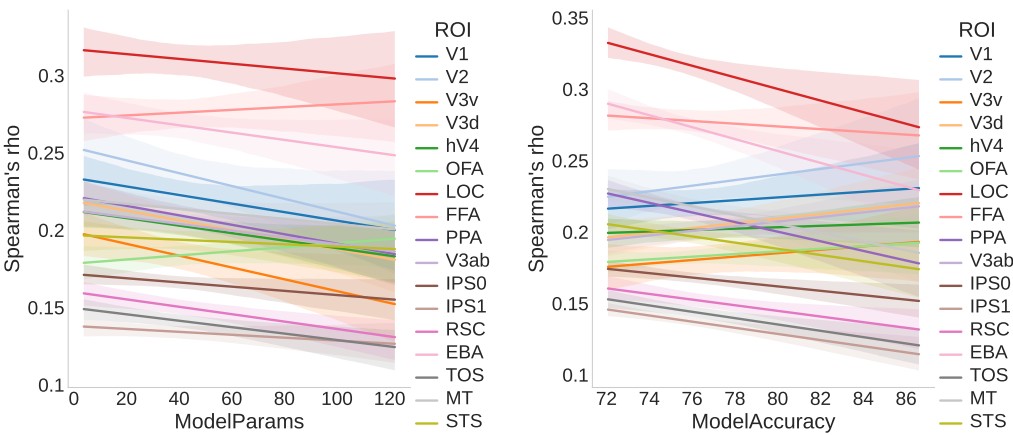

Figure 21: Similar to Figure 6a but with model parameters and model accuracy. Here the trends are less consistent than with the model FLOPs.

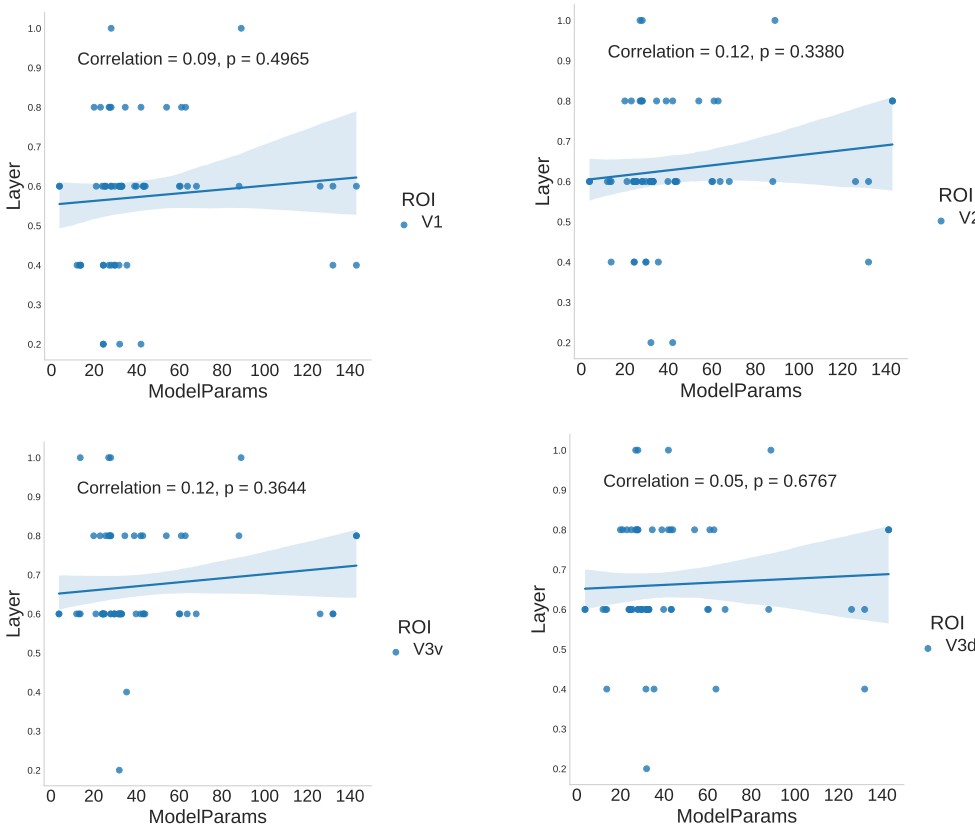

Figure 22: Relation of model parameters with the top-aligned layer to the Early Visual Cortex in (all) CNNs. Early-layer correlation does not seem to be a function of model size.

