# OpenReview forum: "One Hundred Neural Networks and Brains Watching Videos: Lessons from Alignment"
_ICLR.cc/2025/Conference — ICLR 2025 Poster_

### Official Review · Reviewer_YhDR · 2024-10-31

**Soundness:** 3
**Presentation:** 3
**Contribution:** 3
**Rating:** 6
**Confidence:** 4

**Summary:**

This paper presents a comprehensive benchmarking study that compares deep video models with human brain responses to natural video stimuli. The authors utilize a video-fMRI dataset to investigate how various factors, such as temporal modeling, classification tasks, architecture, and training datasets, influence the alignment between neural network representations and human visual processing. The study reveals some insights into the mechanisms of video processing in both artificial and biological systems.

**Strengths:**

The paper addresses a novel and important question regarding the alignment of neural network representations with human brain activity in the context of video processing. The approach of systematically benchmarking a large number of models is innovative and contributes to the understanding of how different architectures and training datasets affect representational alignment.

The paper is clearly written and well-organized, making complex concepts accessible to readers. The figures and tables effectively illustrate the findings, aiding in the communication of key results.

**Weaknesses:**

The temporal resolution of the fMRI dataset. As mentioned by the authors, the low temporal resolution of fMRI is insufficient for representing the brain's processing of temporal information. It is recommended to consider using brain imaging data with higher temporal resolution, such as MEG, for this comparative study.

The diversity of video stimuli data is lacking. Although the entire video database contains 1102 samples, the authors only used 102 for analysis. Why not use all of them? Since each video clip is only 3 seconds long and scene transitions are very limited, using only 102 samples makes it difficult to draw universally applicable conclusions.

As mentioned by the authors, similar studies comparing brain responses to model representations on the natural scenes have been published in previous work. Although this paper tests a richer set of models, the novelty of its method and conclusions are relatively limited.

The authors mention that this work can inspire us to design more efficient video processing models. How exactly do these comparisons motivate the design of more efficient video processing models? Based on the experimental results, it would be meaningful if the authors could give some reasonable suggestions for designing more efficient video models.

Is there any project code (such as a GitHub link) to ensure the reproducibility of the experiments?

**Questions:**

The diversity of video stimuli data is lacking. Although the entire video database contains 1102 samples, the authors only used 102 for analysis. Why not use all of them? Since each video clip is only 3 seconds long and scene transitions are very limited, using only 102 samples makes it difficult to draw universally applicable conclusions.

How exactly do these comparisons motivate the design of more efficient video processing models? Based on the experimental results, it would be meaningful if the authors could give some reasonable suggestions for designing more efficient video models.

Is there any project code (such as a GitHub link) to ensure the reproducibility of the experiments?

---

> ### Author Response · Authors · 2024-11-22
>
> Thank you for acknowledging our systematic approach, contributions, and clear presentation, as well as pointing out important weaknesses in a constructive way. We have revised the manuscript and uploaded the new version with all changes relative to the original shown in blue. Our responses to your points one by one are as follows.
>
> ### Weakness 1: The temporal resolution of the fMRI dataset. As mentioned by the authors, the low temporal resolution of fMRI is insufficient for representing the brain's processing of temporal information. It is recommended to consider using brain imaging data with higher temporal resolution, such as MEG, for this comparative study.
> Thank you for bringing this up - we agree that imaging data with higher temporal resolution such as MEG from humans watching videos, or in combination with fMRI, might be more suitable for this task. This, to the best of our knowledge, does not exist yet (at least publicly). Nevertheless, even with the low temporal resolution of fMRI, this first comparative study of video models still manages to uncover differences in the models performing temporal modeling, setting better foundations for a future MEG study. We clarify this more in the updated version of our discussion section.
>
> ### Weakness 2: The diversity of video stimuli data is lacking. Although the entire video database contains 1102 samples, the authors only used 102 for analysis. Why not use all of them? Since each video clip is only 3 seconds long and scene transitions are very limited, using only 102 samples makes it difficult to draw universally applicable conclusions.
> Please see answer below, in the question version of this point.
>
> ### Weakness 3: As mentioned by the authors, similar studies comparing brain responses to model representations on the natural scenes have been published in previous work. Although this paper tests a richer set of models, the novelty of its method and conclusions are relatively limited.
> Thank you for this comment. We are aware that our methods originate from previous work on alignment in the domain of brain responses to static images. However, here we apply these to the new domain of brain responses to video, with video AI models. Obtaining the results that the established methods produce in a new domain is crucial to check their validity - in the video domain, none of the conclusions made for static images had been validated before, and it is not possible to study the influence of temporal modeling in responses to static images. After establishing a line of work for the video domain (e.g. also an MEG/EEG comparative study with the same established methods), the field can move to new alignment methods that are maybe also specific to video (e.g. a form of dynamic alignment for multiple timesteps), which would be a methodology novelty.
>
> ### Weakness 4: The authors mention that this work can inspire us to design more efficient video processing models. How exactly do these comparisons motivate the design of more efficient video processing models? Based on the experimental results, it would be meaningful if the authors could give some reasonable suggestions for designing more efficient video models.
> Please see answer below, in the question version of this point.
>
> ### Weakness 5: Is there any project code (such as a GitHub link) to ensure the reproducibility of the experiments?
> See below, in the question version of this point.

---

> ### Author Response · Authors · 2024-11-22
>
> ### Question 1: The diversity of video stimuli data is lacking. Although the entire video database contains 1102 samples, the authors only used 102 for analysis. Why not use all of them? Since each video clip is only 3 seconds long and scene transitions are very limited, using only 102 samples makes it difficult to draw universally applicable conclusions.
> Thank you for making this point. The reason for choosing only the test set of BMD is that, for RSA (our chosen method of alignment), a higher number of repetitions of the same condition (stimulus) is required to reduce the measurement noise by averaging across them for each subject. The training set has 3 repetitions and the test set 10 repetitions, with the training dataset being specifically collected for the purpose of fitting voxelwise encoding models, which is a different alignment method. Even if we were using voxel-wise encoding, the training set would only be used to train the reweighting of the models’ features to improve brain predictivity, and the brain scores on which the models would be compared would still be on the 102 test set.
> To showcase this point (increased measurement noise with less repetitions), we have now included a plot in the Appendix section D (Figure 17)  with the RSA scores of image models on the training set (due to  computational constraints, running PCA on the video models for 1000 videos would take either too much memory or too much time). It is observed that the pattern of results for the 1000 training videos (differences between image-object and image-action models) is almost identical to the results with the test set, but with a big difference in the overall scale of alignment which is almost halved. We can see that this is mainly due to the increased noise in the measurement and not because of the wider variety of stimuli, because the noise ceilings also similarly drop - so it is not only harder for the models to match the brains, but for the brains to match each other, even including the target subject - this is the Upper Noise Ceiling (UNC). A sentence explaining these results is now also added under section 3.3.
>
> ### Question 2: How exactly do these comparisons motivate the design of more efficient video processing models? Based on the experimental results, it would be meaningful if the authors could give some reasonable suggestions for designing more efficient video models.
> Thank you for the question - here are some initial ideas that we have on efficient design driven by these results. First, we would select the top-aligned models (with low FLOPs) as starting points. Then we can perform extensive ablations on which components specifically drive alignment - some of these model versions might be more efficient and more aligned.
> Second, an idea is to prune the model to the maximum point where the human alignment can be preserved. Test accuracy would also be evaluated to estimate its decrease as a result of pruning, and also compared with other pruning / efficient methods. One possible outcome is that more brain-aligned models could allow for more pruning without losing test accuracy.
> We have added these ideas in the discussion section of the paper.
>
> ### Question 3: Is there any project code (such as a GitHub link) to ensure the reproducibility of the experiments?
> We will get back to you with an anonymous link or zip in the next few days, as the workload for this rebuttal did not yet allow us to write clear instructions for reproducing the conda environment and to verify that the code runs smoothly on a new setup.
>
> Thank you again, for your valuable suggestions. We look forward to your feedback and further discussions.

---

> > ### Author Response · Authors · 2024-11-25
> > **Uploaded project code in Supplementary Material**
> >
> > We have now uploaded our anonymized project code as a zip in the Supplementary Material of the paper.
> > After downloading, please follow the instructions outlined in the README, and if you have any questions or issues let us know.
> > The code will also be released in a public github repo upon publication.

---

> > ### Comment · Reviewer_YhDR · 2024-11-27
> >
> > Thank you for your response. In general, I still feel that the comparison made with only 102 samples has limited reliability and generalization. I maintain my score.

---

### Official Review · Reviewer_Ggym · 2024-11-01

**Soundness:** 3
**Presentation:** 3
**Contribution:** 3
**Rating:** 8
**Confidence:** 3

**Summary:**

This paper investigates representation alignment between the human visual system and task-driven deep learning models in viewing dynamic natural stimuli, measured in representational similarity across brain regions and model layers. The authors conducted a large-scale ablation on a range of vision models and training conditions, including static vs dynamic models, training data composition, and model architectures and sizes. They showed that dynamic models are more aligned with human early visual areas, moreover, convolution-based models exhibit the hierarchical structure of the biological visual system more so than Transformer-based models.

**Strengths:**

- The question of how similar the biological visual system and the machine learning counterpart process visual information, especially in dynamic settings, is of high interest to both the machine learning and computational neuroscience communities.
- Overall the paper is well written. In particular, the Related Work section succinctly defines where/how this work complements existing literature.
- The choice of models and experiments conducted are thorough, and the analysis of representation similarities between model layers and brain regions is more detailed than previous in this area. The findings on CNN-based models appear to exhibit the hierarchical structure while Transformer-based models have similar representation across depth, aligning with previous work in comparing CNN vs ViT internal representation [1]. These results can potentially guide the visual response modeling community on the choice of model architecture for different regions in the visual system.

[1] Raghu, Maithra, et al. "Do vision transformers see like convolutional neural networks?." Advances in neural information processing systems 34 (2021): 12116-12128.

**Weaknesses:**

I don’t have any major concerns with the paper but have some minor questions and suggestions, please see below.

**Questions:**

- Figure 2a shows that image models trained on object classification are (slightly) more aligned to the early and intermediate (V3ab and hV4) visual areas than image models that are trained on action recognition, but are worse in later visual areas. I wonder if the authors have any suggestions as to why this is the case.
- The brackets and asterisk above the box plots in Figure 2a need some spacing between pairs of boxplots. Right now they are all on the same line and it is difficult to read which pair is which.
- The authors reported a negative correlation between models’ FLOPs (computational complexity) and alignment to higher visual areas. Does the same trend hold if we compare the model performance, maybe the validation or test accuracy (or whatever metric that was used) of the model in their respective datasets? e.g. Do more performant models in their specific task lead to worsened alignment?
- The results show that CNNs exhibit a better hierarchy overall, while Transformers are more aligned to early visual areas. I know that ConViT is already part of the model zoo but have the authors examined architectures that use both convolution and self-attention? Such as CvT [2].

[2] Wu, Haiping, et al. "Cvt: Introducing convolutions to vision transformers." Proceedings of the IEEE/CVF international conference on computer vision. 2021.

---

> ### Author Response · Authors · 2024-11-22
>
> Thank you for the positive review acknowledging our contributions and quality of writing, along with suggesting literature and posing questions that concretely improve the paper. Below are the responses to your specific points; we have also uploaded an updated version of the manuscript, with changes relative to the original shown in blue.
>
> ### Question 1: Figure 2a shows that image models trained on object classification are (slightly) more aligned to the early and intermediate (V3ab and hV4) visual areas than image models that are trained on action recognition, but are worse in later visual areas. I wonder if the authors have any suggestions as to why this is the case.
> Thank you for this question. For the image-object models being slightly more aligned than image-action models in early/intermediate regions, this could reflect that the action recognition task is not important in early regions; the action models are first pretrained on ImageNet the same way as the image-object models, and thus the lower alignment must be due to to their finetuning on action recognition, potentially by degrading well-aligning low-/intermediate level representations. In late areas the image-action models are more aligned, suggesting that the action recognition fine-tuning benefits alignment. We have added this remark in the discussion section of the paper.
>
> ### Question 2: The brackets and asterisk above the box plots in Figure 2a need some spacing between pairs of boxplots. Right now they are all on the same line and it is difficult to read which pair is which.
> Thank you for this suggestion which improves the readability of the figure. We adjusted figure 2a with some spacing between the brackets of different pairs of boxplots, so that they are not all on the same line.
>
> ### Question 3: The authors reported a negative correlation between models’ FLOPs (computational complexity) and alignment to higher visual areas. Does the same trend hold if we compare the model performance, maybe the validation or test accuracy (or whatever metric that was used) of the model in their respective datasets? e.g. Do more performant models in their specific task lead to worsened alignment?
> Thank you for the question. We had included the relation to model test accuracy (this was for action recognition models, on Kinetics 400) in Appendix section E, as the correlations were not consistently negative or positive across brain regions. They are mostly positive in earlier processing regions and mostly negative in later regions, but in most regions the correlation was not significant. So the consistent negative relation between FLOPS and alignment is not mirrored in performance. We note this in the final sentence of section 4.
>
> ### Question 4: The results show that CNNs exhibit a better hierarchy overall, while Transformers are more aligned to early visual areas. I know that ConViT is already part of the model zoo but have the authors examined architectures that use both convolution and self-attention? Such as CvT [2].
> Thank you for the suggestion to investigate hybrid models (Transformers using both convolution and self-attention). We have added a new figure in Appendix D (Figure 15) to explore this.
> In the interest of staying within the bounds of the model libraries already utilized in the paper, we examine ConViT and a video transformer, Uniformer, that combines convolution and self-attention.
> We observe that the alignment progression across layers in ConViT is very similar to the other image transformers in all ROIs (subfigure a), but more interestingly, the alignment progression in Uniformer seems to lie somewhat in between that of other video transformers and video CNNs (subfigure b). This is an intriguing insight that will be good to investigate further (with more hybrid models) in future work - we mention this in our updated discussion section.
>
> Lastly, we thank you for highlighting a paper that reflects our findings (paper 1 from your reference list), which we have now incorporated in the discussion, and again thank you for the rest of your valuable recommendations.

---

> > ### Comment · Reviewer_Ggym · 2024-11-26
> >
> > It is interesting that CvT is similar to other transformers but ConViT sits somewhere between CNNs and transformers even though both are hybrid models. Future work that ablates different components in these hybrid models can help us better understand what made the CNN-based models replicate the overall hierarchy of the visual system. At the same time, transformers have higher representation alignment with the early visual area.
> >
> > I thank the authors for their detailed responses which have addressed my main questions and concerns.

---

### Official Review · Reviewer_gi1E · 2024-11-03

**Soundness:** 2
**Presentation:** 2
**Contribution:** 3
**Rating:** 6
**Confidence:** 4

**Summary:**

The paper aims to address an important question, which is the correlation between human brain activations to complex stimuli - video-watching - recorded with fMRI and extracted features from various deep learning models. The authors compared about 100 different deep learning models (image-based, video-based) for ten different subjects. They used different criteria to evaluate the correlation between brain and deep learning models' activations: 1. temporal modelling (image-based vs video-based) 2. classification tasks (image vs video classification); 3. type of architecture (CNN vs Transformer-based models) 4. training datasets 5. computational complexity of the models. The study identifies various trends, such as the correlation between models' activations and early or high-level regions from the visual cortex, as well as differences between architectures (CNN vs Transformers) and a negative global correlation with models' complexity.

**Strengths:**

- The paper discusses a very interesting and relevant topic. Such a large-scale benchmark, which compares very different and high-performing CNNs and Transformer architectures, is of high value for investigating how image and video models compare with brain feature extraction and hierarchical modelling of visual information.

- The methodology is relatively well-thought-out, with many differentiating criteria to evaluate the impact of important properties of computer vision models, which should process visual information differently: image-based and video-based modelling, different training datasets, transformers vs. CNNs, etc. There are all relevant, and the paper does a good job at analysing the results.

- Statistical tests are carefully run to validate the relevance of the results, which is a great plus to support the validity of the results for such a large-scale study.

- The discussion underlines important points about the role of different layers (in terms of depth), and about the difference in modelling visual features with self-attention (transformer) and convolutions (CNNs)

**Weaknesses:**

- The abstract and introduction lack clarity and could benefit from better writing. For instance, the concepts of "brain representational alignment" and "model-brain alignment" in the abstract are unclear and can be confusing. The second bullet point int= the contributions is not very clear.

- Although the question addressed is highly important/interesting, the reason for conducting such research is not sufficiently well-motivated. Particularly, it is difficult to clearly understand the neuroscience motivations for comparing brain activations to features extracted from deep learning models and how the study would benefit both the development of neuroscience and deep learning architectures in the future.

- One of the main weaknesses is the lack of description of the data acquisition and data preprocessing, which are crucial for a paper of that scale. More extensive (even in the appendix) information about the fMRI acquisition (TR, resolutions, etc.), the preprocessing strategies, and the anatomical/functional registration strategies would be required to fully evaluate how brain signals could be reliably used for comparison. Otherwise, this would undermine the quality of the results and conclusions that can be drawn (see questions below)

- Some points of the methodology could benefit from more clarity (see questions below)

- Some limitations of the study could be better discussed: choice of the fMRI dataset, and the limitations of using the Brain Representational Alignment to model brain alignment.

**Questions:**

**Questions about the limitations of the study**:

- The paper discusses alternatives in the literature to study brain alignment via projecting feature space into voxel space. Still, it should be compared with more recent approaches than those cited in the paper and notably very successful methods that correlated brain activations and language modelling (speech/text), such as [1,2,3]. Is there any reason why those methods are performing in NLP, but maybe are not in image-based/video-based models?

**Methods/Results**:

- I understand the complexity of using high-res feature dimensionality, but it would have been nice to know/see if PCA impact the prediction accuracy (line 188) and the following results.

- How is the feature extraction achieved for the transformer-based architectures (line 270)? This is not clear at the moment.

- Line 275: "Image models trained on action recognition expect the input all at once according to their preprocessing function similar to video models." The temporal modelling for the image-based models on video datasets is not clear. Could you please clarify?

- Could you confirm that the points drawn in Figure 2.a and similar correspond to the maximum activation across layers for any given model? Would it be interesting to study the variance of correlation across layers for each model? Are there any models where layers are completely decorrelated to brain activations or is the correlation generally homogenous across layers? I am not sure this information is currently provided in the current graphs.

- From the graphs in Appendix B, notably Figure 7, it seems that the layers the most responsive to V1/V2 are the mid-layer forms of most of the architectures; what is the rationale, as we know that low-level visual features are extracted by early layers in CNNs based architectures?

- Given that many results indicate that the highest correlation is for the classification layer, would it be interesting to use an unsupervised algorithm? how would you expect the results to differ?

- From Figure 7, it seems that the early visual cortices v1, v2, and v3 have very similar patterns, although one would expect hierarchical modelling in these particular layers. Is this the case? Is there any rational behind?

**Data**

- A thorough description of the dataset used is clearly missing. In particular, what is the voxel resolution, the MRI acquisition parameters, the TR of fMRI?

- Is the brain solely anatomically aligned or functionally aligned? One would expect the brains to be functionally aligned, e.g. using MSM registration [4], as the study compares brain functional activation between different subjects, and anatomical registration might not be sufficient to compare brain activations at the voxel level.

- Are the brains aligned across repetition of the same stimuli? It is not clear if this occurs during the same session or in different sessions. As voxels are averaged across subjects, this seems important.

- Information is missing about whether the study takes into account the temporal lag between brain activation and stimuli to compare brain activation and features. Also, the limitations of using BOLD signals to model brain processing should be mentioned.

- Are the ROIs defined subject-wise or using a template? This should greatly impact the correspondence of functional regions between subjects.

- There is no discussion on the limit of using voxel-wise modelling compared to surface-based modelling. However, surface-based modelling has been shown to be a better predictor of brain function [5,6].


[1] Toward a realistic model of speech processing in the brain with self-supervised learning, J.Millet et al, Neurips 2022

[2] Natural speech reveals the semantic maps that tile human cerebral cortex, AG Huth et al, Nature 2016

[3] Semantic reconstruction of continuous language from non-invasive brain recordings, J Tang et al, Nature Neuroscience 2023

[4] Multimodal surface matching with higher-order smoothness constraints, EC Robinson et al, Neuroimage 2016

[5] A multi-modal parcellation of human cerebral cortex, MF Glasser et al, Nature 2016

[6] The impact of traditional neuroimaging methods on the spatial localization of cortical areas, T Coalson et al, PNAS 2018

---

> ### Author Response · Authors · 2024-11-22
>
> Thank you for your thorough review and actionable feedback. We have revised the manuscript and uploaded the new version with all changes relative to the original shown in blue. Our responses to your points one by one are as follows.
>
> ### Weakness 1: The abstract and introduction lack clarity and could benefit from better writing. For instance, the concepts of "brain representational alignment" and "model-brain alignment" in the abstract are unclear and can be confusing. The second bullet point in the contributions is not very clear.
> Thank you for pointing out our presentation in the abstract and introduction could be improved.
> In the abstract, we changed “brain representational alignment” to “representational alignment to the human brain” and “factors of variation that affect model-brain alignment” to “factors of variation in the models that affect alignment to the brain”. The second bullet point in the contributions now reads “We decouple the alignment effects of temporal modelling from those of action space optimization by adding image action recognition models as control, as well as examine the impact of model architecture and training dataset, all comparing across a fine-grained variety of brain regions.”. We went through the rest of the Abstract and Introduction, leading to a few more repairs that improve clarity.
>
> ### Weakness 2: Although the question addressed is highly important/interesting, the reason for conducting such research is not sufficiently well-motivated. Particularly, it is difficult to clearly understand the neuroscience motivations for comparing brain activations to features extracted from deep learning models and how the study would benefit both the development of neuroscience and deep learning architectures in the future.
> Thank you for this important feedback - we have adjusted the first paragraph of the Introduction to express the motivations more clearly as follows.
>  “Representational alignment is a cornerstone of cognitive computational neuroscience \citep{kriegeskorte2018cognitive}, a discipline that aims to identify neural mechanisms underlying cognition by employing task-performing computational models, such as deep neural networks, for hypothesis testing. This has been shown to be a highly progressive research programme yielding novel neuroscientific insights \citep{doerig2023neuroconnectionist}. Identifying model design choices that strongly impact alignment can not only shed light on the underlying brain mechanisms, but also guide machine learning on borrowing brain's advantages such as efficiency and robustness, for example by using the top brain-aligned designs as starting points for further model development.”
>
> ### Weakness 3: One of the main weaknesses is the lack of description of the data acquisition and data preprocessing, which are crucial for a paper of that scale. More extensive (even in the appendix) information about the fMRI acquisition (TR, resolutions, etc.), the preprocessing strategies, and the anatomical/functional registration strategies would be required to fully evaluate how brain signals could be reliably used for comparison. Otherwise, this would undermine the quality of the results and conclusions that can be drawn (see questions below)
> We now provide a comprehensive summary of the fMRI data acquisition and preprocessing in Appendix section A (see also specific answers to your questions below).
>
> ### Weakness 4: Some points of the methodology could benefit from more clarity (see questions below)
> Thank you for highlighting these points - we refer you to the specific answers to your questions below.
>
> ### Weakness 5: Some limitations of the study could be better discussed: choice of the fMRI dataset, and the limitations of using the Brain Representational Alignment to model brain alignment.
> Thank you for this comment - our limitations section now starts with
> “We base our choice of alignment metric on the comparison of RSA and veRSA made in \cite{conwell2022can}, but this was on static images and it could be worthwhile to make the same comparison for our video data, as well as comparison between other metrics. RSA does not examine possible gains in predictivity as a result of linear feature reweighting.
> [...] Our results are based on a single fMRI dataset, and not validated across multiple fMRI experimental setups.”

---

> ### Author Response · Authors · 2024-11-22
>
> ### Question about the limitations: The paper discusses alternatives in the literature to study brain alignment via projecting feature space into voxel space. Still, it should be compared with more recent approaches than those cited in the paper and notably very successful methods that correlated brain activations and language modelling (speech/text), such as [1,2,3]. Is there any reason why those methods are performing in NLP, but maybe are not in image-based/video-based models?
> Thank you for your question and the opportunity to clarify this point. We do agree that voxel-wise encoding is also a very successful method in NLP as well as in vision studies.
> We have updated this section to include the domain of speech as well as more recent studies.
> We would furthermore like to clarify that by choosing RSA, we are not implying that voxel-wise encoding is not a successful or well-performing method. Rather, we believe that the choice between the two depends on two factors bound to the purpose of each study:
>
> 1. Performing a univariate vs. a multivariate analysis (with respect to activation arrays of brain voxels in specific ROIs, or model features)
> 2. Measuring the stricter emergent (out-of-the-box) model alignment to the brain vs. the more flexible (with the freedom to linearly re-weight model features) maximum achievable brain predictivity score of the model.
>
> For studies such as reference 3 from your list, which is a reconstruction/decoding study, having a voxel-wise encoding approach is crucial as the goal is to predict the activations of the decoded sentences as accurately as possible. In this study, we were interested in a ROI-targeted, multivariate analysis and, based on our intent to perform an out-of-the-box benchmarking of AI models, we wanted the alignment measure to be stricter rather than more flexible. Having these first RSA findings (with the stricter metric) as foundations, future work can move to explore gains in predictivity as a result of feature reweighting with voxelwise modeling.  We have slightly rephrased our wording in section 3.1 to further clarify this.
>
> ### Methods Question 1: I understand the complexity of using high-res feature dimensionality, but it would have been nice to know/see if PCA impact the prediction accuracy (line 188) and the following results.
> Thank you for this important remark. We have now included a figure in Appendix section D (Figure 16) that compares the results with no dimensionality reduction, with SRP, and with PCA.  We believe the inclusion of this analysis considerably strengthens the soundness of the paper’s methodology. Briefly, this additional analysis can be summarized as follows:
> 1. Early Visual Cortex (EVC) alignment scores with full dimensions are lower than with PCA for all 3 model types, and all model layer depths. Especially for image models (object & action), values are much lower and more so in the middle layers, and so they fail to display the early-late hierarchy.
> 2. In higher-level brain regions, scores with full dimensions are more similar to PCA in the top-aligned layer (subfigure a). This is because the top-aligned layer is the last (classification) layer, which is low-dimensional anyway, so the effects of large original dimensionality do not apply. In the progression across all layers (subfigure b), scores in the early and middle layers are again lower with full dimensions than with PCA for all 3 model types.
>
> Thus, overall, the models' RDMs match the brain less when they are constructed with full dimensions than when keeping important components. The same happens with SRP, which is a randomized choice of dimensions, still required to be large in order to be accurate. We believe that results with full dimensions / SRP can be misleading, as for example in (1) they make it seem as if the video models capture the hierarchy better, but in reality the image (object & action) model values are lower than they can be with keeping the important components. Therefore we report only the PCA reduced results in the main text.
>
> ### Methods Question 2: How is the feature extraction achieved for the transformer-based architectures (line 270)? This is not clear at the moment.
> Both Transformers and CNNs are treated similarly in the feature extraction. The extraction is performed with a torch extractor object that places hooks on the desired layers to be extracted. After extraction, layers’ features for all image patches are flattened, producing a single one-dimensional feature vector per layer. We have updated section 3.2 accordingly.

---

> > ### Author Response · Authors · 2024-11-22
> >
> > ### Methods Question 3: Line 275: "Image models trained on action recognition expect the input all at once according to their preprocessing function similar to video models." The temporal modelling for the image-based models on video datasets is not clear. Could you please clarify?
> > For image-based models trained on video datasets, there is only trivial temporal modelling - as outlined in the previous paragraph “Model choice”, we consider as trivial anything that either makes completely separate computations per frame and only averages frame features before classification, or aggregates frames with static pooling operations at different stages.
> > However, in order to perform these trivial operations, the models are still trained (and thus also evaluated) with the whole video as a single input sample, and so are fed the whole video instead of single frames. We agree the sentence you pointed out was unclear, so we have updated it to read “Action recognition image models were trained on sequences of frames as input samples (only to aggregate with trivial pooling operations), so for those we perform inference on video inputs, as we do for video models”.
> >
> > ### Methods Question 4: Could you confirm that the points drawn in Figure 2.a and similar correspond to the maximum activation across layers for any given model? Would it be interesting to study the variance of correlation across layers for each model? Are there any models where layers are completely decorrelated to brain activations or is the correlation generally homogenous across layers? I am not sure this information is currently provided in the current graphs.
> > Points in figure 2a and similar indeed correspond to the maximum alignment (not activation) across layers. It is definitely important to also express the variance of correlation (alignment scores) across layers, and we do this by plotting separate scores for each layer using their mean and standard deviation within a given model family (figure 2b and similar). However, we agree it is also interesting to report these plots for each model separately to uncover potential outliers in the layer progression pattern. Thus, we have included an additional figure in Appendix section D (Figure 14) . We observe that there are no noticeable outliers (with respect to the mean layer pattern) in the Early Visual Cortex (EVC), but there are a few in later areas. Specifically, the maximum alignment (peak) around depth=0.2 in all late regions corresponds to the two SlowFast models, the peak around depth=0.6 to the CSN models, and the models that remain very low-scoring until depth=0.8 are variants of VideoSwin and TimeSformer. There are several potential explanations for these interesting differences in model alignment across layers; for example, the early peak for the SlowFast models could be due to the dual-stream architecture which could allow earlier computation of more semantic features. Future work can address this in more detail, for example through ablations on the model components.
> >
> > ### Methods Question 5: From the graphs in Appendix B, notably Figure 7, it seems that the layers the most responsive to V1/V2 are the mid-layer forms of most of the architectures; what is the rationale, as we know that low-level visual features are extracted by early layers in CNNs based architectures?
> > When comparing the representations of Transformers and CNNs, it has been shown that the same representations computed in layers 0-30 of a ViT are computed in layers 0-60 in ResNet50 [b]. This could be a similar effect, where CNNs compute low-level features for more layers of the network. This is especially true for deep architectures, which make up most of the architectures we consider in this study. The discussion section has been updated to reflect this finding.
> >
> > [b] Raghu, Maithra, et al. "Do vision transformers see like convolutional neural networks?." Advances in Neural Information Processing Systems 34 (2021): 12116-12128.
> >
> > ### Methods Question 6: Given that many results indicate that the highest correlation is for the classification layer, would it be interesting to use an unsupervised algorithm? how would you expect the results to differ?
> > We have added a figure in Appendix section D (Figure 18) comparing VideoMAE_B, which is pretrained in a self-supervised manner and is included in the library mmaction2, with and without (videomae_B_pretrainONLY) supervised fine-tuning (on Kinetics 400). It seems that in all ROIs, the supervised fine-tuning is crucial for alignment, although the alignment of the pretrainONLY model is still significantly larger than zero. However, we would like to stress that we cannot draw a general conclusion for all self-supervised models from this one example, and a future study could focus specifically on this aspect.

---

> ### Author Response · Authors · 2024-11-22
>
> ### Methods Question 7: From Figure 7, it seems that the early visual cortices v1, v2, and v3 have very similar patterns, although one would expect hierarchical modelling in these particular layers. Is this the case? Is there any rational behind?
> It is true that V1 and V2 look pretty similar, however V3 noticeably differs - it shows an upwards tilt in the layer lines for all 3 model types (i.e. the alignment at depth = 1 is now greater than the alignment at depth = 0). This trend continues going into the intermediate regions hV4, V3ab, with the pattern there being in between Early Visual Cortex and late areas.
> As to why V1 and V2 are very similar, this could be related to the measurement or alignment method not discerning the nuances of the representations in these areas.
>
> ### Data Question 1: A thorough description of the dataset used is clearly missing. In particular, what is the voxel resolution, the MRI acquisition parameters, the TR of fMRI?
> We have now added a summary of the dataset, including these relevant parameters, in Appendix A: MRI data acquisition. All details of the publicly available dataset can be found in Lahner et al. 2024.
>
> ### Data Question 2: Is the brain solely anatomically aligned or functionally aligned? One would expect the brains to be functionally aligned, e.g. using MSM registration [4], as the study compares brain functional activation between different subjects, and anatomical registration might not be sufficient to compare brain activations at the voxel level.
> We use a preprocessed version of the MRI data provided by the original authors of the BMD (Lahner et al., 2024). This preprocessing was conducted using the standard fMRIprep pipeline, which does not include advanced methods such as MSM. Following standard practice in fMRI studies, functional MRI scans containing BOLD responses to the videos were aligned to anatomical reference scans and then to standard MNI space. This last step is necessary for the application of ROIs derived from standard atlases to each individual subject’s BOLD data (see further discussion below). Importantly, however, the RSA approach does not entail any averaging or direct comparison of voxels across subjects. Instead, voxel activations for each individual subject and ROI form subject- and ROI-specific RDMs which are separately correlated with model RDMs as described in section 3.1 of our paper. We now clarify these details of the dataset in section Appendix A: MRI data preprocessing and Appendix A: ROI definitions.
>
> ### Data Question 3: Are the brains aligned across repetition of the same stimuli? It is not clear if this occurs during the same session or in different sessions. As voxels are averaged across subjects, this seems important.
> As noted above, voxels are not averaged across subjects, but rather RSA scores are computed separately for each subject and then averaged across subjects. However, the voxels of each subject are indeed averaged across multiple repeats of the same stimuli. This is described in section 3.1 of our paper. These repeats of the same stimuli were randomly distributed across 4 sessions (sessions 2-5), necessitating alignment of runs between sessions. This is achieved by fMRIPrep by alignment to the anatomical reference and then standard MNI space (see above).
>
> ### Data Question 4: Information is missing about whether the study takes into account the temporal lag between brain activation and stimuli to compare brain activation and features. Also, the limitations of using BOLD signals to model brain processing should be mentioned.
> Temporal lag is modeled in the General Linear Model that is used to estimate the brain activation to each stimulus (referred to as beta estimates) by including a hemodynamic response function (HRF). The preprocessing version from Lahner et al., (2024) that we used employs a state-of-the-art GLM method referred to as GLMsingle (Prince et al., 2022), which fits an optimal HRF for each voxel. We now include these details and relevant references in Appendix A: Brain response estimation.
> As for the limitations of using BOLD signals to model brain processing, we adjusted our limitations section to state “Next, BOLD signals are indirect measurements of brain activity, which in turn is an indirect measurement of the brain's representations - these are latent variables that cannot be measured.”

---

> ### Author Response · Authors · 2024-11-22
>
> ### Data Question 5: Are the ROIs defined subject-wise or using a template? This should greatly impact the correspondence of functional regions between subjects.
> ROIs were defined using templates from three commonly used atlases, namely the Wang atlas for retinotopic ROIs, the Julian atlas (for category-selective ROIs) and the Glasser atlas (your reference 5). Importantly however, the voxels within each of these ROI masks were additionally subjected to a selection criterion based on each individual’s response in the independent functional localizer scans (50% top active voxels), thus also taking into account subject-wise variation. We now describe this in the Appendix A: ROI definitions.
>
> ### Data Question 6: There is no discussion on the limit of using voxel-wise modelling compared to surface-based modelling. However, surface-based modelling has been shown to be a better predictor of brain function [5,6].
> Thank you for the suggestion, we added in the limitations section the following “Surface based modelling could also be considered as a potentially better predictor of brain function \citep{glasser2016multi, coalson2018impact}.”
>
> Finally, we would like to express our gratitude to you for providing this very constructive feedback (and taking the time to do so). We hope that with these changes your opinion of the paper can be improved, and we look forward to your answer and further discussions.

---

> > ### Comment · Reviewer_gi1E · 2024-11-26
> >
> > Thank you very much for your detailed response. I truly appreciate the authors' commitment to reply to all the questions thoroughly. I believe that the additions made by the authors actually improve the manuscript in terms of soundness and results. In particular, thank you for your precision regarding the temporal lag, PCA results, and feature extraction. The description of the data acquisition and processing in the Appendix is very extensive and welcome.
> >
> > Despite the additions in the discussion, particularly in the paragraph comparing the differences in layer activations between CNNs and Transformers (lines 502-505), the neuroscience results and discussion remain somewhat limited. For example, the rationale for early layers being less correlated than mid-layers to early visual systems is not fully explored and discussed (which could be due to limitations of the dataset and/or the experimental setup). Therefore, it is difficult to appreciate all the results from a neuroscience perspective and how all these results would inform our understanding of brain encoding/decoding of visual stimuli or would impact the way deep learning models should be designed.
> >
> > However, this new version is a very good improvement. I would still like to raise my mark from 5 to 6.

---

> > > ### Author Response · Authors · 2024-11-27
> > > **Our thanks to reviewer gi1E**
> > >
> > > Thank you for updating your score, for the appreciation of our work and updated manuscript, as well as for your continued constructive comments.
> > >
> > > We agree that the paragraph you pointed out in the discussion could indeed be enhanced with more (potential) explanations on the finding of mid-depth alignment peak in the CNNs. As you said however, to reach a concrete neuroscientific insight on this, any hypothesis would need to be validated with the same analyses on more video fMRI datasets.
> > >
> > > For completeness, we have added the lines 499-502:
> > > *"Why is the top-aligned layer mid-depth in CNNs, and not earlier? It does not appear to be a matter of network size or depth (see Appendices D, E) - therefore, an interesting hypothesis is that this relates to the dynamic nature of the stimuli, as in image fMRI studies it was not observed \citep{nonaka2021brain}. This hypothesis requires further validation on (multiple) other video fMRI datasets."*
> > >
> > > The above refers to two additional figures in the appendix (Fig. 20, Fig. 22) which rule out the first hypothesis that came to mind on this, namely that shallow or small CNNs would have an earlier instead of a mid-depth peak.

---

### Official Review · Reviewer_dMT3 · 2024-11-05

**Soundness:** 3
**Presentation:** 4
**Contribution:** 3
**Rating:** 8
**Confidence:** 1

**Summary:**

This paper investigates the roles of temporal modeling and action recognition optimization in aligning video models with brain activity patterns during video observation. The findings show that early visual processing regions in the brain benefit from temporal modeling, aligning with neuroscience literature that these areas are sensitive to short-term changes, whereas late areas are more semantically oriented and responsive to action categories. Comparisons between CNNs and Transformers reveal that Transformers align with early brain regions at shallower layers due to their global attention mechanism, suggesting a different spatial processing approach than CNNs, which makes sense when compared with our intuitions about the models. The study also finds that computationally efficient models align better with high-level brain regions, implying that achieving human-like abstract semantics does not require high computational resources. Limitations include reliance on indirect brain measurements, fMRI's low temporal resolution, and a limited set of available video models.

**Strengths:**

- Interesting paper, great exploration, well written.
- The study provides a nuanced understanding of how temporal modeling and action recognition optimization affect alignment with different stages of brain processing, offering valuable insights for both neuroscience and machine learning.
- By comparing CNNs and Transformers, the paper highlights unique alignment patterns, suggesting that global attention mechanisms in Transformers may capture early brain processing more effectively than CNNs.
- The finding that computationally efficient models align better with high-level brain areas supports the development of optimized models that achieve human-like abstract representations without excessive computational costs.
- The research bridges machine learning and neuroscience, potentially guiding the design of video models that better align with human brain activity (particularly in applications involving temporal and semantic processing).

**Weaknesses:**

- One significant limitation is the narrow focus on visual regions, neglecting other brain networks that are also engaged during video watching. Motion and imagery cues in videos can activate regions beyond visual areas, including those related to emotion, memory, and multisensory integration. This raises the question of why the study did not analyze the entire brain, as doing so could potentially yield more comprehensive insights and improve alignment accuracy by leveraging richer information from other relevant brain networks.
- The study only benchmarks models available in a specific library, limiting comparisons across diverse training paradigms (e.g., supervised, contrastive, self-supervised). This restricts the generalizability of findings and leaves open questions about whether models trained with different methods would exhibit distinct alignment patterns.

**Questions:**

- Why limit the analysis to visual regions only? Motion and imagery cues in videos can activate brain networks beyond the visual cortex.
- There are several recent visual/neural decoding papers (such as MindEye, MindVis, etc.). How do they compare numerically with what is presented in this paper?

---

> ### Author Response · Authors · 2024-11-22
>
> Thank you for the positive review acknowledging our contributions and quality of writing, as well as outlining limitations and posing interesting questions. Below are the responses to your specific points; we have also uploaded an updated version of the manuscript, with changes relative to the original shown in blue.
> ### Weakness 1: "One significant limitation is the narrow focus on visual regions, neglecting other brain networks that are also engaged during video watching. Motion and imagery cues in videos can activate regions beyond visual areas, including those related to emotion, memory, and multisensory integration. This raises the question of why the study did not analyze the entire brain, as doing so could potentially yield more comprehensive insights and improve alignment accuracy by leveraging richer information from other relevant brain networks."
> Studying other brain regions engaged during video watching, such as those related to emotion or memory, sounds indeed very promising. The reason why this study focused only on the visual system, was because we wanted to set the baseline for visual areas responding to videos first, in line with the previous work extensively comparing neural networks to these areas on static image stimuli (e.g. Conwell et al., 2022). At the same time, to study other regions we would need their ROI mapping which was not included in the BMD dataset, or otherwise move to non-ROI based analysis - this also points to exploring this in a future study, rather than this one. We have added this promising research opportunity to our future work discussion.
>
> ### Weakness 2: The study only benchmarks models available in a specific library, limiting comparisons across diverse training paradigms (e.g., supervised, contrastive, self-supervised). This restricts the generalizability of findings and leaves open questions about whether models trained with different methods would exhibit distinct alignment patterns.
> We agree that comparing different training objectives is an important question to address in benchmarking studies such as ours. However, here we chose to use specific libraries in order to make the study more self-contained and reproducible, and easier to systematically control one factor at a time. Outside of time limitations, one could sample models from all recent and highly cited publications uniformly to ask the intended kinds of questions. Specifically for the different learning paradigms, this could also be a study by itself that we will highlight in future work.
>
> ### Question 1: Why limit the analysis to visual regions only? Motion and imagery cues in videos can activate brain networks beyond the visual cortex.
> Please see our response to the “weakness” version of this point above.
>
> ### Question 2: There are several recent visual/neural decoding papers (such as MindEye, MindVis, etc.). How do they compare numerically with what is presented in this paper?
> While the focus of such papers is usually on achieving optimal reconstructions/decoding of the sensory inputs, and less on identifying differences in alignment for specific brain regions, these papers sometimes also report results on the numerical contributions of brain regions in the reconstruction. Depending on what is reconstructed (semantics or structure), the contributions of brain regions could also vary, similarly to the alignment of different model types (image-object, image-action, video-action). The closest decoding study to ours would be one reconstructing video from fMRI, such as [a]. In figure 7 of this paper, the authors plot the contributions of each brain region to the reconstruction of the semantics, structure, or motion of the videos, and find, similarly to our study, that intermediate and late regions are more connected to video semantics, and early regions are more connected to the low-level structure, as well as the motion of the videos. We now incorporated this comparison in our discussion section.
>
> [a] Lu, Yizhuo, et al. "Animate Your Thoughts: Decoupled Reconstruction of Dynamic Natural Vision from Slow Brain Activity." arXiv preprint arXiv:2405.03280 (2024).
>
> Thank you again for the valuable suggestions.

---

> > ### Comment · Reviewer_dMT3 · 2024-11-26
> >
> > Thank you for the responses! I will keep my score!

---

### Meta-Review · Area_Chair_s1oW · 2024-12-18

**Metareview:**

In this paper, the authors presented a benchmark study of deep learning video models on representational alignment to human brain (vision region), for temporal modelling by viewing dynamic natural video stimuli. Specifically, based on a public dataset, this study compared 99 image and video models (including both CNNs and Transformers) and 10 human subjects' brain (fMRI) data, measured by representational similarity across brain regions and layers in deep models. The findings from different perspectives showed the correlations between models' activations and visual cortex, architectural differences, and other findings that align with neuroscience literature. The strengths of this paper include:
- The investigation of representational alignment between deep models and the human brain visual system, in a dynamic setting, is an important problem that could offer insights and value to both machine learning and computational neuroscience communities.
- The presented large-scale benchmark study covered a wide range of deep models and different settings. The corresponding comparison and analysis at this scale is a good contribution and valuable to following research in this direction.
- The experimental findings are interesting, especially between different network architectures, and the hierarchical alignment properties. These findings, some aligned with neuroscience literature and other related works, could provide good guidance for the community on model choice.
- The paper is well-written and well-organised, providing a good reading experience to both communities.

The main weaknesses of this paper are around: 1) the limited direct implications and contributions/insights for neuroscience, i.e. not many new findings were reported about how the human brain processes vision information; 2) the limited samples in the chosen dataset; 3) the narrow focus only on the visual regions; and 4) lack of technical novelty due to the commonly used RSA method.

Considering the above, although this study indeed is limited in several perspectives and the findings might not be that generalisable, given the scale and variety of the analysed deep models, both the AC and reviewers still found it to be valuable to the community, and could be a good start and inspire following research in this direction. As a result, the AC is happy to recommend an Accept.

**Additional Comments On Reviewer Discussion:**

This paper received long back-and-forth discussions during the rebuttal and discussion phases (including both authors-reviewers and AC-reviewers). The authors provided a detailed rebuttal to the reviewers' comments, which addressed most of the major concerns raised by the reviewers, as acknowledged by the reviewers' responses. Further questions and discussions were followed, clearing more concerns. Although there are still remaining concerns about the limitations of this study, during the AC-reviewers discussion, reviewers agreed on the value of this paper to the community, and were happy to see this paper being accepted.

Overall, this paper received a consistent positive final rating (of 2 Accept and 2 borderline Accept). The AC agreed with the comments and ratings from the reviewers and supported the acceptance of this paper.

---

### Decision · Program_Chairs · 2025-01-22

Accept (Poster)